# Thermal Comfort and Energy Performance of Atrium in Mediterranean Climate

**Reihaneh Aram * and Halil Zafer Alibaba *** 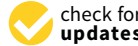

Faculty of Architecture, Department of Architecture, Eastern Mediterranean University,
99628 Gazimagusa, Northern Cyprus

* Correspondence: aireen1371@gmail.com (R.A.); halil.alibaba@emu.edu.tr (H.Z.A.);
  Tel.: +90-533-856-7001 (R.A.); +90-533-863-0881 (H.Z.A.)

**Abstract:** This paper aims to determine the optimal single-story office building model with a corner atrium regarding different atrium orientations and office-building window-opening ratios in the Mediterranean climate via EDSL Tas software. When window-opening ratios were 25% and 50% at the northeast and southeast orientations of atriums and office spaces, thermal comfort was achieved according to categories B and C, respectively, within the cold season. Additionally, for the northeast atrium orientation with 25%, 137.2 W and 189.5 W of heat loss and gain in the office zone, and 37.7 W and 204.7 W of heat loss and gain in the atrium zone were recorded. Moreover, for the northeast atrium orientation with 50%, 134.5 W and 134.2 W of heat loss and gain in the office zone, and 40 W and 192 W of heat loss and gain in the atrium zone were recorded. On the other hand, for the southeast atrium orientation with 25%, 108.7 W and 143 W of heat loss and gain in the office zone, and 68.8 W and 130 W of heat loss and gain in the atrium zone were recorded, while, with 50%, 111.7 W and 142.7 W of heat loss and gain in the office zone, and 67.5 W and 121.2 W of heat loss and gain in the atrium zone were recorded. In the warm season, the atrium and office spaces were not thermally comfortable.

**Keywords:** thermal comfort; energy performance; atrium; EDSL Tas; predicted mean vote (PMV); predicted percentage of dissatisfied (PPD)

## 1. Introduction

As an important issue in buildings for the provision of occupant's comfort in the different seasons, indoor thermal comfort cannot just rely on passive strategies of cooling. So, for determining building performance during different seasons, dynamic thermal simulations can be used in finding the optimal model for specific designs and climates. Therefore, analyzing the building's thermal comfort and building energy performance are important parameters used in academic research. Generally concerns over energy efficiency have improved considerably over recent years. According to the European Union (EU), it is economically and technically feasible to produce different progressive strategies in all active sectors. Accordingly, the building sector is one of the most important sectors for improving energy efficiency [1]. So applying passive strategies can be sufficient solution like as transitional spaces as atrium.

The user's comfort and energy efficiency are two important primary concerns for assessing the performance of building system controls. According to the California Commercial End-Use study [2,3], the primary uses of electricity in buildings include: interior lighting, ventilation, refrigeration and cooling. Furthermore, natural gas is used for water heating, space heating, and cooking. Overall, lighting, ventilation, air conditioning (HVAC) and heating systems account for 60% of the electric power consumed by buildings, while the rest of the usage is attributed to other kinds of equipment

depending on the building's function. However, because occupants spend about 80% of their lives inside the building, lighting and HVAC systems are vital for providing a comfortable environment for the occupants. As a result, these systems are also important for improving productivity, which depends on the occupants' visual comfort, thermal comfort and indoor air quality comfort [3,4]. Although the definition of thermal comfort is presented as "the condition of the mind which expresses the satisfaction with the thermal environment" [5], it is not defined only as a mental condition. Thermal comfort is best understood as being based also on environmental physical parameters in addition to the effects of the cultural, psychological and social dimensions [6]. Thermal comfort has been found to have a direct relationship with the inhabitants' thermal sensation, which is influenced by certain environmental parameters such as air temperature, air movement, mean radiant temperature and humidity. However, the users' activities as well as their choice of clothing are also effective factors [7,8]. Providing thermal comfort is a key effective parameter in building design. A significant achievement of HVAC (heating, ventilation and air conditioning) systems is that they generate suitable thermal comfort for users. The effective and important variables for thermal comfort include the surrounding air temperature, radiant temperature around a person, air velocity above the person, the surrounding air humidity, the persons' activities and the persons' clothes [9].

As an important point, meteorological parameters are some of the important factors in electricity usage which influence office buildings' energy performance. Additionally, extra energy is required in maintaining an adequate working space; for instance, in creating a comfortable working environment, it is vital to remove excess heat [10]. Therefore, a common problem encountered in office buildings is the excessive use of machinery to ensure users' comfort [11]. Designing buildings with low energy consumption may also be used to achieve the occupants' thermal comfort due to the changing environmental conditions. Accordingly, radiant and cooling systems may be combined with other practical low energy strategies and adaptive systems to reduce the energy consumption of buildings, as well as carbon dioxide emissions [12]. The growing awareness concerning energy efficiency has drawn the attention of designers to the atrium area as a potential way to control energy misuse and damage to the natural environment by diffusing excess harmful gases. Consequently, this may be used as a method for providing thermal comfort, while at the same time reducing energy consumption [13]. A suitable environmental thermal comfort has a direct effect on the occupants' health and their activities in the building. However, the necessity of indoor thermal comfort may increase the building's energy consumption [14]. The Fanger framework established the PMV index, which is the predicted mean vote, and the PPD index as the predicted percentage of dissatisfied. The PMV index includes multiple factors, such as air velocity, air temperature, globe temperature, relative humidity and metabolic rate [15].

In a hot climate, the envelope and openings such as windows have an important and effective role in the building's energy savings. Indeed, the use of clear glass may increase the thermal transmittance in a good way (2.00 W/m$^2$K and 3.00 W/m$^2$K) [16,17]. In controlling thermal comfort, operating windows have a direct connection to the physical parameters on the outside and natural ventilation. Furthermore, it can reduce the cooling loads while improving the air quality inside at the same time [10,11]. The window opening ratio would be based on the outside air temperature, time, the user's pattern and season [17]. So, it can be summarized that the window opening ratio is an extremely important factor in regards to the users' indoor comfort and energy consumption.

The atrium is one of the most famous and common transitional spaces in recent years. Atriums consist of an enormous glass wall and roof, and due to the high transmittance of the glass material for heat and solar absorptance, the atriums' indoor thermal space is strongly affected by outdoor environmental conditions [12]. Atriums are increasingly being incorporated in a variety of building types due to their potential for success in social, environmental and economic aspects. As can be seen in Figure 1, atriums allow daylight to penetrate into the deep parts of the building's plan [18], improve manufacturing costs and energy efficiency in the building sector and cause a positive psychological effect on the users. Furthermore, atriums have been introduced as a common feature of contemporary

buildings, especially in tall buildings, as they have various advantages, such as improving the visual links between the indoor and outdoor spaces as part of the building, creating a powerful focal point, creating spaciousness, stimulating the environment, and serving an iconic role [19]. Additionally, large and centrally glazed spaces have been a popular area for centuries, since the ancient times starting in Mesopotamia [8]. As seen in Figure 2, atriums are classified into nine categories based on the physical dimensions of width (W), length (L) and height (H), and the geometrical ratio defined as: a = VW, b = H/W and c = H/L. Categories V to IX are square-like and the lowest form [19]. The atrium which was analyzed in this study was the partial length atrium type with the typical dimensions.

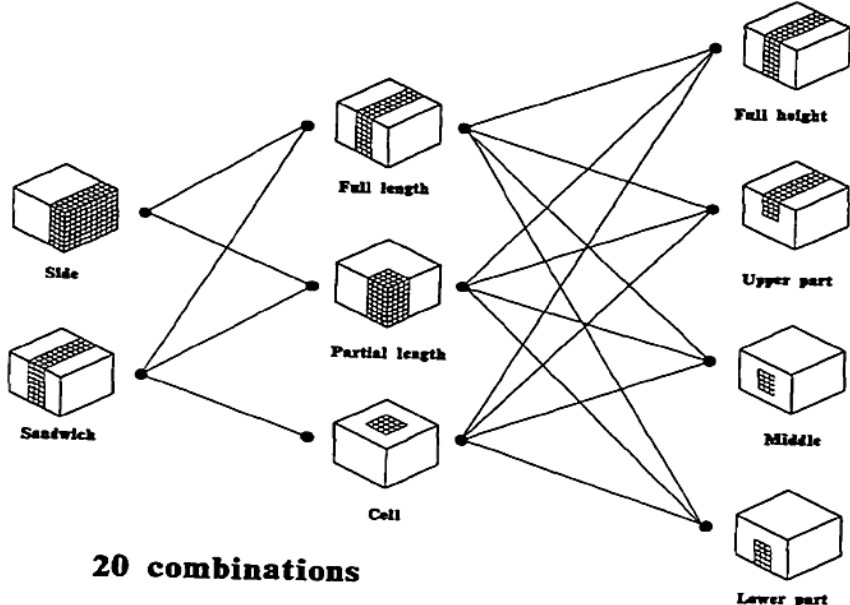

**Figure 1.** Atrium orientation in buildings [19].

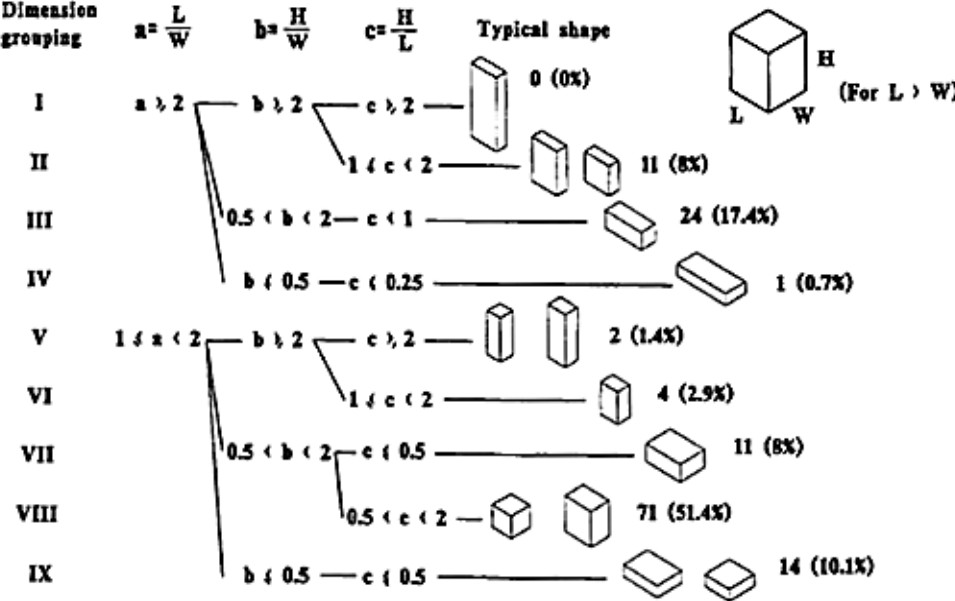

**Figure 2.** The geometrical classification of atriums [19].

The shape of atriums is a key parameter in transmitting daylight, and distributing the light into adjoining spaces is a difficult issue [20,21]. In designing a suitable atrium with natural ventilation, the relevant design factors may be divided into two different categories, which include the "parameters

influencing thermal performance" and "parameters influencing ventilation performance." Furthermore, assessing the thermal performance of the atrium has been achieved using various methods, such as analytical, experimental, numerical and mathematical modelling in different studies [14]. Kim and Boyer proposed a set of physical models, where these scale models defined the direct relationship of the atrium and its shape, such as linear, square and rectangular shapes, to the DF (daylight factor) as the center of the open atrium. Furthermore, the reflectance values of the walls and the floor were fixed as 0.3 and 0.1, respectively. Subsequently, the DF of the center of the atrium and the floor was calculated as: DFcf = 117e − 0.996WI [18,22].

## 2. Objective

The aim of this paper is to determine the optimal single-story office building model with a corner atrium type regarding different atrium orientations and office building window opening ratios in the Mediterranean climate. In this study, all simulation models were prepared via the EDSL (environmental design solutions limited) Tas software [23] using 0%, 25%, 50%, 75% and 100% opening ratios for all windows, and four main atrium orientations in a single-story office building as north-west, north-east, south-west and south-east to analyze thermal comfort and building energy performance based on the ASHRAE 55, 2013–ISO 7730: 2005 and EN 15251: 2007 standards [6,24,25]. The energy performance and thermal comfort parameters were analyzed over the course of one year. Furthermore, acceptable averages of a predicted mean vote (PMV) and a predicted percentage of dissatisfied (PPD) were applied for this study. According to the ISO 7730 [24] and EN 15251 [25] standards, there is a number of categories used to determine thermal comfort. These categories are: category A where PPD < 6% and PMV ranges from −0.20 to 0.20; category B where PPD < 10% and PMV ranges from −0.50 to 0.50; category C where PPD < 15% and PMV ranges from −0.70 to 0.70, and category D which is inclusive of the minimum value start of category C.

## 3. Literature Review

According to the global internal comfort targets adopted by developing countries, there has been a greater focus on thermo-hydrometric parameters leading to the development of smart building devices and passive strategy systems [26]. Nowadays, the question of energy efficiency has resulted in serious demands and environmental concerns, which have drawn the attention of building designers to natural ventilation in summer, the use of daylight as a source of energy for buildings and using solar heat in winter. The result of this advanced technological design is an integrated glazing atrium, which can enhance all the environmental parameters in indoor spaces [27]. An overall building design can be improved by integrating appropriate energy-efficient features, including those in the building layout and façade. As such, the building design contributes to cost-effective energy usage and overall energy efficiency in terms of: architectural design through passive strategies, including the building's orientation, building's elements of insulation, glazing, the atrium's cooling and heating function, reflective glazing, passive solar system and landscaping, while designing electrical and mechanical systems using active energy management strategies [28].

Studies on atriums' energy performance have been carried out since 1980. For instance, in 1987, a skylight handbook and the design for energy conservation with skylights reported that atrium fenestration could reduce energy usage for lighting and cooling [29,30]. Various studies investigating atrium energy performance have identified the important design elements of the atrium which significantly affect a building's energy performance. These elements include the atrium's geometry, glazing, ventilation system and shading configuration. Obviously, the interior environmental parameters also play important roles in assessing parameters, such as thermal comfort, daylighting, air quality and ventilation based on the building elements, which also include the envelope, orientation, HVAC system, roof aperture, roof transmittance, glazing area, attribution of the adjacent spaces, and atrium type and shape [30].

Historically, the atrium has been used as a place for increasing light in indoor spaces since the nineteenth century [31,32]. 'Atrium' is originally a Latin word referring to the central court or room in which the walls are covered in black soot. While this area was referred to as the atrium in a typical ancient Roman house, the term in modern architecture now refers to a new form of space with glass walls and roof, which create connected areas throughout the building [8]. As mentioned in other studies, one main problem that is faced with the atrium as a box-shaped glass building is providing comfortable conditions in a hot and humid climate, for instance, by controlling excessive daylight, glare and temperature [33,34].

Moreover, atriums have some disadvantages in terms of energy consumption such as gaining a large amount of solar heat in summer and air stratification, which directly affects the atrium's performance and users' comfort. In the light of these points, passive strategies for energy saving that may be used include providing a buffer zone between the indoor and outdoor (atrium's buffer zone causes complicated thermal phenomena), introducing natural light into the users' space and natural ventilation (usable for energy efficiency without the installation of air conditioner systems) [35]. Previous studies on buoyancy-driven air ventilation reported that, in a single-story building, it is not vital to increase the atrium's height to improve the ventilation performance of the space that connects with the atrium, and the outlet configuration of an atrium has a remarkable effect on the stack impact. Additionally, other effective parameters in atrium buildings include buoyancy-driven natural ventilation, the building's geometry, the stack openings' location, and the distribution of the heat resources [36].

Various studies conducted on thermal elaboration prediction have utilized a single index based on the Fanger model, which initially proposed PMV (predicted mean vote) [37]. Furthermore, the PMV model can generate a more accurate prediction of air temperature, natural ventilation and comfort [8,37]. For physical measurements, the Fanger model uses the concept of heat balance in the theory of a human body to derive four effective environmental parameters for human thermal comfort which are air velocity (m/s), air temperature (°C), relative humidity (RH), and the mean radiant temperature (MRT) [37,38]. In assessing building energy consumption, there are certain vital evaluation criteria, such as cooling, heating, room acoustics, ventilation, life cycle and lighting; hygro-thermal evaluation criteria such as density, emissivity, conductivity, solar absorption, vapor diffusion and specific heat; criteria related to photo-colorimetry consisting of reflectance, surface roughness and specularity; and criteria related to LCIA (life cycle impact assessment), which include life service, material and fabrication impacts, as well as maintenance [39].

The ratio of the atrium construction has an impact on annual energy consumption, and by reducing this ratio, it is possible to decrease the annual energy consumption. For example, a study reported the minimum and maximum annual energy consumption ratios as 1:2 and 1:10, respectively. Moreover, the thermal comfort conditions of these ratios were close to the standards of Fanger's PMV model [32].

A study investigating atrium energy consumption found that the atrium's space in a five-story building in Santiago had more than 75% savings in its cooling energy consumption, which is also known to be affected by the building's function, atrium type and type of glass [40,41]. The atrium building typology has a heating average in the winter season which is higher than that in the courtyard typology. This situation suggests that, in hot-humid and hot-dry climates, this transitional area can decrease the heating demand for the atrium building, in spite of its potential for overheating in summer, which increases the ratio of discomfort hours [42].

As a passive strategy, which is mentioned above, the atrium as a transitional space establishes a connection with the outdoor space; consequently, the atrium is converting daylight, which directly affects the building energy performance. Using daylight via the atrium space can improve building energy efficiency, but only by controlling the amount of daylight in the internal spaces according to the occupants' comfort [43]. However, using shading devices can have advantages in a tropical climate

because it causes decreases in the inner surface temperature of the indoor space and also decreases the atrium's heat gain, although this situation generates discomfort spaces in the upper levels [44].

In a cold climate, another parameter that directly affects thermal comfort and energy performance is atrium geometry. For instance, the problem with shallow atrium types is that during the summer time the atrium space becomes overheated in the vertical and horizontal directions, but throughout the winter time, this turns into an advantage [30].

The revolution in building design based on sustainability has allowed the integration of passive strategies to enable sufficient building energy performance levels [45]. With respect to previous studies, it appears vital to simultaneously consider the atrium in approaches to users' thermal comfort and energy performance without using shading devices in finding the net assessment result. Accordingly, to determine the optimal single office building with corner atrium type, it is vital to have the optimal parameters as window opening ratios in all building spaces and atrium placement based on the climate of Gazimagusa, Cyprus. It is clear that atriums cause different problems in hot and humid climates throughout the year, and these problems include overheating in summer time and heating losses in winter time. Consequently, ensuring users' comfort requires the extensive use of energy for mechanical systems. This tendency makes it necessary to determine the suitable parameters of the atrium in a hot and humid climate. Also, the findings of this research can be used as practical information, which explains the amount of changes in thermal comfort and energy performance in a single story building for different atrium orientations and all window opening ratios.

## 4. Methodology

*Dynamic Thermal Simulations and Analysis*

In the first part of the research, the building was simulated in Northern Cyprus, which may be classified as a hot and humid climate based on the weather data collected for Gazimagusa, CYP (34.88° North, 33.63° East). In the Gazimagusa climate, from June to September has the highest average temperature (29 °C–32 °C), while the minimum average temperature during the warm seasons is about 23 °C. The cold season is from December to March (6 °C–19 °C), and the minimum average is 7 °C. Generally, the warmest month is August with an average temperature of 26.6 °C and the coldest month is January with an average temperature of 11.8 °C. Furthermore, January is the most (80% humidity) and June (55% humidity) is the least humid month during the year. This weather data information was obtained from the nearest weather station to Gazimagusa, CYP [46]. The maximum and minimum heat losses and gains of atrium with office spaces are shown in Figure 5. The monthly PMV and PPD for the atrium and office were analyzed according to different categories in Figures 6–21. The achieved thermal comfort depending on different categories of PPD, PMV, draught, vertical air temperature, cool or warm floor, and radiant temperature asymmetry are compared for different months in Tables 1–6.

The simulation and analysis were carried out using the EDSL Tas software version 9.3.3 [23]. The office model consisted of a single-story building with a square-shaped (15 m × 15 m), open planning and partial-length atrium (5 m × 5 m). The model analysis was conducted for four different simulation model groups of the atrium orientation and building window opening ratios of 0%, 25%, 50%, 75% and 100% as shown in Figures 3 and 4. The PMV (predicted mean vote) and PPD (percentage predicted dissatisfied) analysis process was conducted for office hours from 9 am–6 pm without weekends. All of the simulation models were shaped as a partial length atrium on the corner side of the building with the atrium roof window opening as the central part and individual wall window openings. Moreover, the space of the office zone was an open plan design with window openings on each wall as shown in Figure 3. The atrium orientation was changed in the office plan according to the north-west, north-east, south-west and south-east directions. Furthermore, all of the window openings in the office space and atrium space were tested using the same opening proportions of 0%, 25%, 50%, 75% and 100%. In the next part, all of the simulation models were re-evaluated according to their atrium orientation, which was moved to different locations in the

building plan. The buildings in the office simulation models had two zones: the office space and the atrium space. Additionally, the PMV parameters included air speed values between 0.15 m/s as the lower limit and 0.3 m/s as the upper limit, metabolic rate was 1.2 met, and the clothing value was 0.6 clo as the lower limit value and 0.95 clo as the upper limit value. The office building schedules were created throughout the whole year. In this study, the energy performance was analyzed using various parameters such as gaining and losing heat flow (W), mean radiant temperature (MRT), space temperature (internal and external temperature) and relative humidity. Furthermore, for investigating thermal comfort, the data for the percentage of people dissatisfied (PPD) and predicted mean vote (PMV) calculations were generated via the EDSL Tas software [23] by using the thermal comfort prediction macros section. Additionally, all of the aforementioned parameters were used in assessing the entirety of the building, i.e., the atrium space and the office space. Figure 4 depicts the simulation and analysis process with details of each stage.

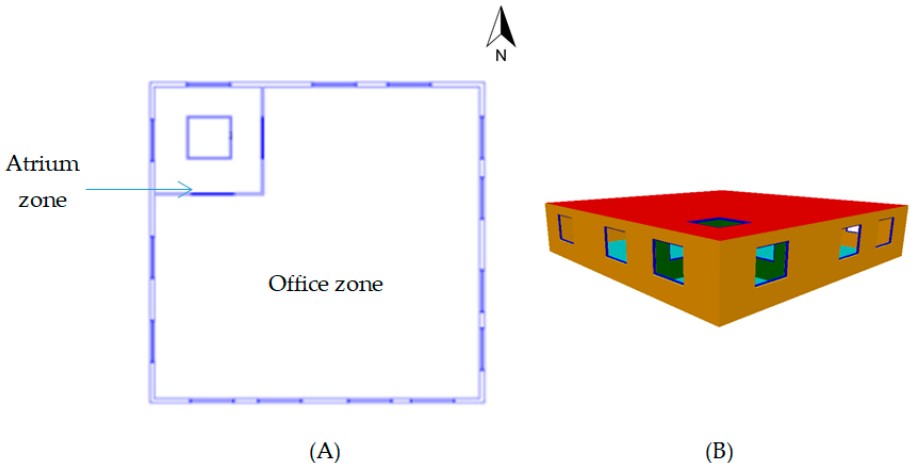

(A)                                                          (B)

**Figure 3.** Case study models, as sample plan (**A**) and three-dimensional (3D) model (**B**) of the flat office building.

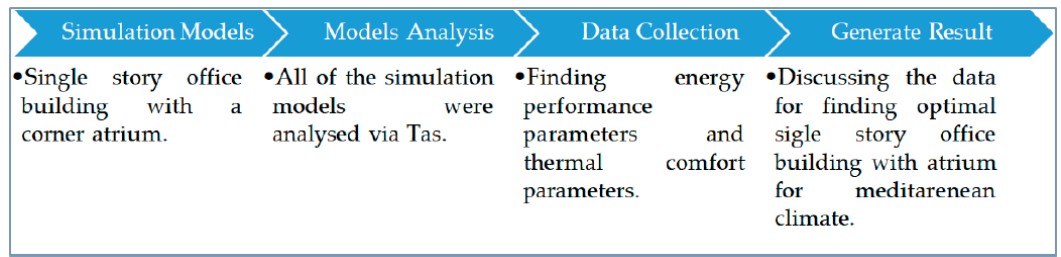

**Figure 4.** The research simulation and analysis process.

- ■ According to the arrangement of thermal environments recommended by ISO 7730 [24] and EN 15251 [25], the following parameters were used:
- ■ Category 1(A), 2(B), 3(C), 4(D) (EN 15251), as Predicted Percentage of Dissatisfied (%), <6, <10, <15, >15.
- ■ Category 1(A), 2(B), 3(C), 4(D) (EN 15251), as Predicted Mean Vote Range, -0.20 to 0.20, -0.50 to 0.50, -0.70 to 0.70, <-0.70 or >0.70.
- ■ Category 1(A), 2(B), 3(C), as Percentage of Dissatisfied (PD) Due to Draught (%), <10, <20, <30.
- ■ Category 1(A), 2(B), 3(C), as PD Due to Vertical Air Temperature, the difference (%), <3, <5, <10.
- ■ Category 1(A), 2(B), 3(C), as PD Due to Cool or Warm Floor (%), <10, <10, <15.
- ■ Category 1(A), 2(B), 3(C), as PD Due to Radiant Temperature Asymmetry (%), <5, <5, <10.
- ■ All of the simulation models of the office building used opaque construction layers with U-values and properties as follows:

- ■ Ground floor: External U-value with horizontal flow direction is 0.283 (W/m² °C), 0.760 external and 0.500 Internal surface of solar absorptance, 0.910 external and 0.900 internal emissivity (W/m² °C), 0.297 conductance (W/m² °C) and 127.999 time constant.
- ■ Ceiling: External U-value with horizontal flow direction is 1.01 (W/m² °C), 0.700 external and 0.500 internal surface of solar absorptance, 0.900 external and internal emissivity (W/m² °C), 1.251 conductance (W/m² °C), 13.749 time constant.
- ■ Brick external walls: 229 mm plastered brick wall with horizontal flow direction is 1.135 (W/m² °C), 0.400 external and internal surface of solar absorptance, 0.900 external and internal emissivity (W/m² °C), 1.407 conductance (W/m² °C), 4.920 time constant.
- ■ All of the simulation models of the office building used glass construction layers with U-values, and properties as follows:
- ■ Glass for all windows (clear 6-12-6 double glazing low E): 1.94 (W/m² °C), 0.498 solar transmittance, 0.550, 0.162 external and 0.157 internal for external solar absorptance, 0.209 external and 0.107 internal for internal solar absorptance, 0.797 light transmittance, 0.845 external and 0.845 internal emissivity, 2.896 conductance (W/m² °C), 0 for time constant with no blind.

## 5. Results and Discussions on Energy Performance

As seen in Figure 5, analyzing the energy performance of the office building simulation models with different atrium orientations and window-opening ratios requires consideration of the building heat transfer and mean radiant temperature of the office and atrium zones. The heat transfer of the southwest atrium orientation in the office zone of the building simulation model with a 0% window-opening ratio had a maximum heat loss of 1448.8 W at 5:00 p.m. on 15 January and a minimum loss of 0.01 W at 4:00 a.m. on 14 May. Additionally, its maximum heat gain was 1768.5 W at 7:00 a.m. on 14 February, with a minimum gain of 0 W at 1:00 a.m. on 1 January. The maximum heat loss in the atrium zone, however, was 446 W at 5:00 p.m. on 15 January with a minimum heat loss of 0.04 W at 3:00 p.m. on 16 November. Furthermore, the maximum heat gain was 717.7 W at 4:00 p.m. on 15 January, while the minimum was 0.03 W at 6:00 a.m. on 30 May. In this group, when the window-opening ratio was increased to 25%, the office zone had a maximum heat loss value of 1448.8 W at 5:00 p.m. on 15 January and a minimum heat loss of 0.01 W at 4:00 a.m. on 17 May. However, this zone had a maximum heat gain value of 1768.5 W at 7:00 a.m. on 14 February and a minimum heat gain value of 1.1 W at 8:00 p.m. on 11 January.

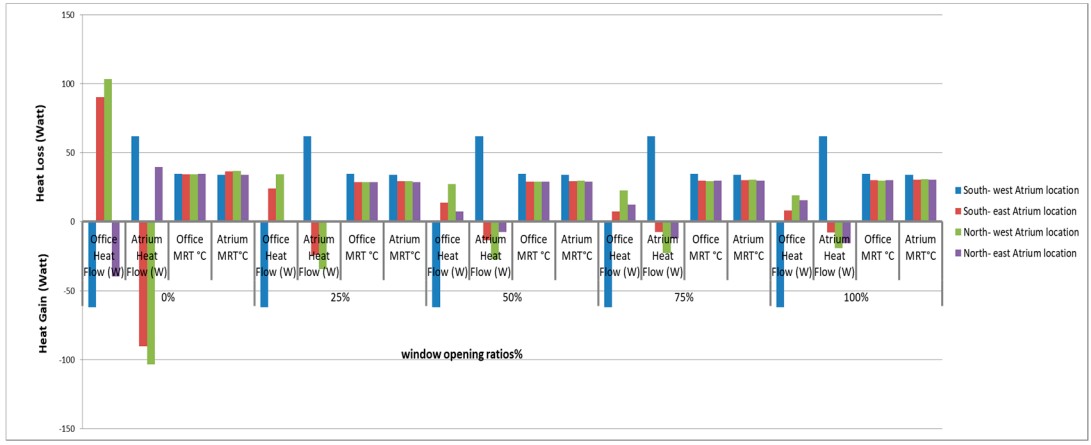

**Figure 5.** The building heat flow (W) and mean radiant temperature (°C) for the office and atrium zones in different atrium orientation simulation models.

The southwest atrium orientation in the atrium zone had a maximum heat loss of 446 W at 5:00 p.m. on 15 January and a minimum value of 0.04 W at 3:00 p.m. on 16 November. Additionally, the maximum heat gain for this zone was 717.7 W at 4:00 p.m. on 15 January and the minimum

gain was 0.03 W at 6:00 a.m. on 30 May. When the window-opening ratio was increased to 50%, the office zone had its maximum heat loss as 1448.8 W at 5:00 p.m. on 15 May, and its minimum as 0.01 W at 4:00 a.m. on 14 May. Furthermore, the maximum heat gain of this zone was 1768 W at 7:00 a.m. on 14 February with the minimum of 0.02 W at 10:00 p.m. on 13 September. The atrium zone had the maximum heat loss value of 446 W at 5:00 p.m. on 15 January, while its minimum value was 0.04 W at 3:00 p.m. on 28 April. The maximum heat gain in the atrium space was 717.7 W at 4:00 p.m. on 15 January and the minimum heat gain was 0.03 W at 6:00 a.m. on 30 May. At the 75% window-opening ratio with the same atrium orientation, the office zone had the maximum heat loss of 1448.8 W at 5:00 p.m. on 15 January, while its minimum was 0.01 W at 4:00 a.m. on 17 May. Additionally, the maximum heat gain of this zone was 1768.5 W at 7:00 a.m. on 14 January and the minimum gain was 0.02 W at 10:00p.m. on 13 September. The atrium zone's maximum heat loss stood at 446 W at 5:00 p.m. on 15 January, while the minimum was 0.04 W at 3:00 p.m. on 16 November. Moreover, the maximum heat gain of the atrium zone was 717.7 W at 4:00 p.m. on 15 January, and the minimum was 0.03 W at 6:00 a.m. on 30 May. Furthermore, when the windows were completely opened (100% window-opening ratio) in this group, the office zone had a 1448.8 W maximum heat loss value at 5:00 p.m. on 15 January and a minimum loss of 0.01 W at 4:00 a.m. on 17 May. Additionally, the heat gain in the office space had the 1768.5 W maximum at 7:00 a.m. on 14 February and the minimum of 0.02 W at 10:00 p.m. on 13 September. The atrium zone had its maximum heat loss value as 446 W at 5:00 p.m. on 15 January, and its minimum was 0.04 W at 3:00 p.m. on 18 November. This zone also had the maximum heat gain of 717.7 W at 4:00 p.m. on 15 January and the minimum heat gain of 0.03 W at 6:00 p.m. on 30 May.

When all of the windows were closed completely (0% window-opening ratio) in the southeast atrium orientation of the office building simulation models, the office zone had a 1526.8 W maximum heat loss at 5:00 p.m. on 15 January and the minimum of 0.08 W heat loss at 3:00 p.m. on 7 May. Additionally, this zone had a 1262.8 W heat gain maximum at 4:00 p.m. on 15 January and the minimum as 0.01 W heat gain at 4:00 a.m. on 29 September. The atrium zone had the maximum heat loss value of 391.5 W at 5:00 p.m. on 15 January, and its minimum was 0.06 W heat loss at 4:00 p.m. on 27 June. Additionally, the atrium zone had the 983.6 W maximum heat gain at 6:00 a.m. on 26 April, and its minimum was 0.1 W heat gain at 2:00 p.m. on 5 May. When the window-opening ratio was increased to 25%, the office zone had a 1206.4 W maximum heat loss at 1:00 a.m. on 1 January and the minimum of 0.01 W at 10:00 a.m. on 6 February. Furthermore, this zone had the 1094.3 W heat gain maximum at 9:00 a.m. on 31 December, and its minimum heat gain was 0.02 W at 7:00 a.m. on 6 June. The atrium zone had the 259.2 W maximum heat loss at 1:00 a.m. on 1 January and the 0.1 W minimum heat loss at 12 a.m. on 25 April. Additionally, the atrium space had the maximum 561 W heat gain at 5:00 a.m. on 19 July and the 0.05 W minimum heat gain at 11:00 a.m. on 16 April.

For the northwest atrium orientation in the office building simulation model, when all the window spaces were closed (0% window opening ratio), the office zone had the 1639.7 W maximum heat loss at 5:00 p.m. on 15 January, and its minimum was 0.2 W heat loss at 5:00 a.m. on 18 September. The maximum heat gain in this space was 1374 W at 6:00 a.m. on 4 April and the minimum gain was 0.05 W at 4:00 a.m. on 27 February. Moreover, when all the windows were closed, the atrium zone had the 289.8 W maximum heat loss at 5:00 p.m. on 15 January and the minimum of 0.01 W at 8:00 p.m. on 1 January. The heat gain in the atrium zone had the maximum value of 1446.8 W at 4:00 p.m. on 15 January and the minimum of 0.05 W at 9:00 p.m. on 25 February. At a 25% window-opening ratio in this group of the simulation model, the office zone had the 1273.1 W maximum heat loss at 1:00 a.m. on 1 January and the minimum of 0 W at 10:00 p.m. on 28 June. The office zone also had the 1255.4 W maximum heat gain at 6:00 a.m. and the minimum heat gain of 0 W at 10:00 p.m. on 28 June. The atrium zone had the 221.8 W maximum heat loss at 6:00 a.m. on 15 March, and its minimum was 0 W at 11:00 p.m. on 20 December. In this space, the maximum heat gain was 667.5 W at 4:00 p.m. on 15 January and the minimum gain was 0 W heat gain at 11:00 p.m. on 20 December. When the window-opening ratio was increased to 50%, the office zone had the 1324.2 W maximum heat loss

at 1:00 a.m. on 1 January and the minimum of 0.01 at 11:00 p.m. on 18 March. However, it had the maximum heat gain value of 1276.5 W at 6:00 a.m. on 5 April and a 0.03 W minimum heat gain at 2:00 a.m. on 27 October.

The northwest atrium orientation in the atrium zone had a 213.8 W maximum heat loss at 6:00 a.m. on 15 March, and its minimum was 0.03 heat loss at 7:00 a.m. on 26 March. Additionally, the atrium space had a 614.5 W maximum heat gain at 4:00 p.m. on 15 January and the minimum of 0.05 W heat gain at 3:00 a.m. on 23 October. At a 75% window-opening ratio, the office zone in this group had a 1340.5 W maximum heat loss at 1:00 a.m. on 1 January and a 0.06 W minimum heat loss at 4:00 a.m. on 28 November. Additionally, the office zone had a 1344 W maximum heat gain at 6:00 a.m. on 5 April, and its minimum was a 0.07 W heat gain at 11:00 a.m. on 22 January. The atrium zone of this group had a 204.8 W maximum heat loss at 6:00 a.m. on 15 March, and its minimum was a 0 W heat loss at 10:00 p.m. on 10 March. In addition, the atrium zone had a 595.5 W maximum heat gain at 4:00 p.m. on 15 January, and its minimum was 0 W heat gain at 10:00 p.m. on 10 March. When all of the windows were opened completely (100% opening), the office zone had a 1349.2 W maximum heat loss at 1:00 a.m. on 1 January and a 0.04 W minimum heat loss at 2:00 a.m. on 7 April. It also had a 1372 W maximum heat gain at 6:00 a.m. on 4 April and the minimum heat gain value of 0.01 W at 5:00 p.m. on 20 December. The atrium zone with the same simulation parameters had a 199.9 W maximum heat loss value at 11:00 p.m. on 25 November and its minimum heat loss value as 0.03 W at 5:00 p.m. on 17 September. Furthermore, the atrium zone had a 634.2 W maximum heat gain at 4:00 p.m. on 15 January and the minimum of 0.03 W heat gain at 6:00 p.m. on 21 January.

The office zone in the northeast atrium orientation of the office building simulation model with a 0% window-opening ratio had a 1667.9 W maximum heat loss value at 5:00 p.m. on 15 January, and its minimum was a 0.06 W heat loss at 6:00 a.m. on 11 November. The office space also had a 719.6 W maximum heat gain at 4:00 p.m. on 15 January, and its minimum was a 0.06 W heat gain at 6:00 a.m. on 31 March. The atrium zone of this simulation group had a 194 W maximum heat loss at 5:00 p.m. on 15 January, and its minimum was a 0 W heat loss at 7:00 p.m. on 27 August. Furthermore, the atrium zone had a 1230.7 W maximum heat gain at 4:00 p.m. on 15 January and the minimum 0 W heat gain at 7:00 p.m. on 27 August. At a 25% window-opening ratio, the office zone of this group had a 1259.5 W maximum heat loss at 1:00 a.m. on 1 January, and its minimum was a 0.03 W heat loss at 4:00 p.m. on 21 July. Additionally, the office zone had a 1074.8 W maximum heat gain value at 3:00 p.m. on 25 January and the minimum heat gain value of 0.02 W at 12:00 a.m. on 4 December. The atrium zone had a 210.4 W maximum heat loss at 1:00 a.m. on 1 January and a minimum of 0.04 W heat loss at 12:00 a.m. on 23 February. Additionally, the atrium space had a 604.6 W maximum heat gain at 7:00 a.m. on 14 February, and its minimum was a 0.01 W heat gain at 12:00 p.m. on 23 February.

This study illustrated that the atrium orientation in the office building plan had a direct effect on the indoor thermal condition and the building's energy performance. With regard to the building's heat flow, the monthly results depicted that the dynamic simulation models with the southwest atrium orientation had similar and sometimes identical heat loss and gain during the year with different window-opening ratios in the office zone and atrium zone, in contrast to the other atrium orientations. Throughout January, February, March, and April, these simulation models had from 190.1 W to 222.8 W heat loss and 191.9 W to 233.9 W heat gain in the office zone. Additionally, they had from 75.63 W to 138.2 W heat loss and 73.2 W to 181.6 W heat gain in the atrium zone. The dynamic simulation models with the southeast atrium orientation during the cold season and the 0% window-opening ratio in the office zone had from 118.9 W up to 163 W heat loss and from 82.4 W up to 167.9 W heat gain; the ones with the 25% window-opening ratio had from 86.9 W to 130.3 W heat loss and 123.6 W to 191.6 W heat gain in same season; the ones with the 50% window-opening ratio had between 91.5 W and 135.7 W heat loss and between 118 W and 190 W heat gain; the ones with the 75% window-opening ratio had 110.8 W to 191.4 W heat loss and 116.4 W to 188.7 W heat gain; and the ones with the 100% window-opening ratio had between 94.5 W and 141.1 W heat loss and between 115.5 W and 134.8 W heat gain.

Furthermore, the southeast atrium orientations during the cold season in the atrium zone when the opening area was completely closed, the 0% window-opening ratio had from 72.1 W up to 135.6 W heat loss and from 161.8 W up to 271 W heat gain; the ones with the 25% window-opening ratio had 56.6 W to 87.2 W heat loss and 100.2 W to 157.8 W heat gain; the ones with the 50% window-opening ratio had 57.5 W to 83.1 W heat loss and 95.4 W to 149.8 W heat gain; the ones with the 75% window-opening ratio had 57.9 W to 81.2 W heat loss and 92.4 W to 147.2 W heat gain; and the ones with the 100% window-opening ratio had 57.8 W to 79 W heat loss and 92 W to 142.4 W heat gain. The office building simulation models with the northwest atrium orientation when the office zone openings were completely closed during the cold season had from 217.3 W to 255 W heat loss and from 257.3 W to 357.7 W heat gain. However, the BHT (building heat flow) heat flow decreased slightly in the office zones with the 25% window-opening ratio, which had 143.6 W to 188 W heat loss and 156.3 W to 247.8 W heat gain; the ones with the 50% window-opening ratio had 139.1 W to 187.6 W heat loss and 158 W to 250 W heat gain; the ones with the 75% window-opening ratio had 138 W to 187.2 W heat loss and 164.2 W to 255.2 W heat gain; and the ones with the 100% window-opening ratio had 139 W to 187.4 W heat loss and 164.3 W to 257.2 W heat gain. Moreover, in this group, the atrium zones with the 0% window-opening ratio had 16.4 W to 26.9 W heat loss and 205.2 W to 262.8 W heat gain; the ones with the 25% window-opening ratio had 44.4 W to 57.9 W heat loss and 144.2 W to 217.4 W heat gain; the ones with the 50% window-opening ratio had 42.8 W to 54.4 W heat loss and 131.6 W to 204.5 W heat gain; the ones with the 75% window-opening ratio had 41.1 W to 52 W heat loss and 128.6 W to 199.4 W heat gain; and the ones with the 100% window-opening ratio had 39.9 W to 51.2 W heat loss and 126.3 W to 192.8 W heat gain.

The office zone in the northeast atrium orientations of the office building simulation model in January, February, March, and April with the 0% window-opening ratio had from 183.6 W up to 234.3 W heat loss and from 87.2 W up to 125.9 W heat gain; the ones with the 25% window-opening ratio had 121.3 W to 153.8 W heat loss and 91.2 W to 168.4 W heat gain; the ones with the 50% window-opening ratio had 116.4 W to 152.4 W heat loss and 99 W to 183.8 W heat gain; the ones with the 75% window-opening ratio had 114.2 W to 152.2 W heat loss and 101.5 W to 192 W heat gain; and, when the office area windows were completely opened (100% opening), these values were 113.4 W to 151.1 W heat loss and 99.2 W to 197.3 W heat gain. Furthermore, this dynamic simulation group in the atrium zone when the atrium area windows were completely closed had from 15.7 W up to 24.2 W heat loss and from 236.9 W up to 306.5 W heat gain; the ones with the 25% window-opening ratio had 29.5 W to 46.5 W heat loss and 173.2 W to 223.9 W heat gain; the ones with the 50% window-opening ratio had 31.9 W to 47 W heat loss and 159.9 W to 210.6 W heat gain; the ones with the 75% window-opening ratio had 32.5 W to 47 W heat loss and 157.2 W to 203.8 W heat gain; and the ones with the 100% window-opening ratio had 32.2 W to 47.5 W heat loss and 156.7 W to 201.8 W heat gain.

During the warm season in May, June, July, and August, the office zone in the southwest atrium orientation simulation models when all the windows were closed had 163.4 W to 225.4 W heat loss and 189.1 W to 321.7 W heat gain; the ones with the 25% window-opening ratio had 163.7 W to 225.6 W heat loss and 188.9 W to 321.7 W heat gain; the ones with the 50% window-opening ratio had 163.7 W to 225.6 W heat loss and 188.9 W to 360.4 W heat gain; the ones with the 75% window-opening ratio had 125.6 W to 163.7 W heat loss and 223.2 W to 360.4 W heat gain; and the ones with the 100% window-opening ratio had 163.7 W to 225.6 W heat loss and 360.4 W heat gain. In the atrium zone in this group, the 0% window-opening ratio had from 34.8 W up to 51.7 W heat loss and from 188.9 W up to 246.3 W heat gain; the ones with the 25% window-opening ratio had 34.8 W to 51.7 W heat loss and 188.9 W to 246.3 W heat gain; the ones with the 50% window-opening ratio had 34.8 W to 43 W heat loss and 237 W to 246.3 W heat gain; the ones with the 75% window-opening ratio had 34.8 W to 43 W heat loss and 188.9 W to 207 W heat gain; and, when atrium space windows were completely opened (100% opening), it had from 34.8 W up to 43 W heat loss and from 188.9 W up to 207.3 W heat gain.

In the same warm season, the southeast atrium orientations in the office zone with a 0% window-opening ratio had from 112.6 W up to 163.3 W heat loss and from 54.5 W up to 99.8 W heat gain; the ones with the 25% window-opening ratio had 115.6 W to 140.3 W heat loss and 94 W to 148.4 W heat gain; the ones with the 50% window opening ratio had 113 W to 138 W heat loss and 94 W to 151.9 W heat gain; the ones with the 75% window-opening ratio had 112 W to 138.1 W heat loss and 92.8 W to 150.8 W heat gain; and the ones with the 100% window-opening ratio had 110.3 W to 138.3 W heat loss and 94.4 W to 151.6 W heat gain. The atrium zone in this dynamic simulation group with a 0% window-opening ratio had from 41.5 W up to 124.5 W heat loss and from 161.3 W up to 274.9W heat gain; the ones with the 25% window-opening ratio had 46.9 W to 72.9 W heat gain; the ones with the 50% window-opening ratio had 48.5 W to 70.8 W heat loss and 81.8 W to 165.4 W heat gain; the ones with the 75% window-opening ratio had 48.3 W to 69.7 W heat loss and 92.4 W to 162.4 W heat gain; and the ones with the 100% window-opening ratio had 47.1 W to 65.4 W heat loss and 75.7 W to 160.4 W heat gain.

The office zone in the office building simulation models with a northeast atrium orientation from May to August and a 0% window-opening ratio had from 163.1 W to 177.8 W heat loss and 87.6 W to 132.5 W heat gain; the ones with the 25% window-opening ratio had 124.7 W to 144.7 W heat loss and 93.9 W to 152.9 W heat gain; the ones with the 50% window-opening ratio had 120.6 W to 139.1 W heat loss and 94.4 W to 160.8 W heat gain; the ones with the 75% window-opening ratio had 119.9 W to 136.7 W heat loss and 93.6 W to 164.5 W heat gain; and the ones with the 100% window-opening ratio had 117.1 W to 136.7 W heat loss and 95.7 W to 161.7 W heat gain. The atrium zone with the same parameters when all windows were closed (0% opening) had from 22.3 W up to 46.7 W heat loss and from 292.2 W up to 348.7 W heat gain; when the window-opening ratio increased to 25%, it had 49.2 W to 71.3 W heat loss and 166.1 W to 182.7 W heat gain; the ones with the 50% window-opening ratio had 47.7 W to 68.5 W heat loss and 158.9 W to 174 W heat gain; the ones with the 75% window-opening ratio had 46.9 W to 67.6 W heat loss and 153.3 W to 171.3 W heat gain; and the ones with the 100% window-opening ratio had from 47 W up to 67.1 W heat loss and from 149.3 W up to 171 W heat gain.

The office zone in the northwest atrium orientation of the simulation building models had fluctuations in heat flow as the 0% window-opening ratio had from 215.6 W up to 264.3 W heat loss and from 222.9 W up to 355.2 W heat gain; the ones with the 25% window-opening ratio had 147.6 W to 173.6 W heat loss and 172.3 W to 254.2 W heat gain; the ones with the 50% window-opening ratio had 144.8 W to 168.8 W heat loss and 175.5 W to 261.6 W heat gain; the ones with the 75% window-opening ratio had 146 W to 163.7 W heat loss and 177.5 W to 259.7 W heat gain; and the ones with the 100% window-opening ratio had 146.1 W to 169.9 W heat loss and 182.6 W to 262.9 W heat gain. The atrium zone in this simulation model group had from 15.1 W up to 23 W heat loss and from 218.6 W up to 229.3 W heat gain; the ones with the 25% window-opening ratio had 45.1 W to 57.3 W heat loss and 166 W to 182.9 W heat gain; the ones with the 50% window-opening ratio had 41 W to 52 W heat loss and 157.9 W to 174.9 W heat gain; the ones with the 75% window-opening ratio had 40.7 W to 49.8 W heat loss and 154.8 W to 168.9 W heat gain; and the ones with the 100% window opening ratio had 41 W to 47.7 W heat loss and 148.8 W to 169.3 W heat gain.

The dynamic simulation models with a southeast atrium orientation in this period and the 0% window-opening ratio had from 180.6 W up to 205.3 W heat loss and from 67.9 W up to 92.1 W heat gain; the ones with the 25% window opening ratio had 131.1 W up to 141.1 W heat loss and 107.9 W up to 164.1 W heat gain; the ones with the 50% window-opening ratio had 128.9 W to 139.7 W heat loss and 118.6 W to 170.6 W heat gain; the ones with the 75% window-opening ratio had 130.1 W to 139.7 W heat loss and 122.9 W to 174.7 W heat gain; and the ones with the 100% window-opening ratio had 131.9 W to 139.7 W heat loss and 126.2 W to 176.8 W heat gain. Furthermore, the atrium zone in this group and the 0% window-opening ratio had from 15.2 W up to 20.3 W heat loss and from 240.3 W up to 304.8 W heat gain; the ones with the 25% window-opening ratio had 30.3 W to 44.5 W heat loss and 146.2 W to 232.1 W heat gain; the ones with the 50% window-opening ratio had 32.8 W to 45.5 W heat loss and 176.7 W to 216.9 W heat gain; the ones with the 75% window-opening ratio had 35 W to

46.4 W heat loss and 168.5 W to 212.5 W heat gain; and the ones with the 100% window-opening ratio had 35 W to 46.6 W heat loss and 168.4 W to 209.2 W heat gain.

## 6. Results and Discussions on Thermal Comfort

While analyzing occupants' comfort, it is vital to assess the predicted mean vote (PMV) and predicted percentage dissatisfied (PPD) values in all dynamic simulation models of the office building zones with different atrium orientations and different window-opening ratios. The simulation model of the office building with a southwest atrium orientation had the office temperature as 24.82 °C, atrium area temperature between 21.78 °C and 17.56 °C, office relative humidity ranging from 41.85% to 42.98%, and atrium space relative humidity between 50.12% and 51.32%. As indicated in Figure 6, the solid horizontal lines depict the maximum points of the categories. Consequently, in the office zone with the southwest atrium orientation, when the window opening ratios were 0%, 25%, 50%, 75%, and 100%, the yearly PMV average was 2.61. The maximum PMV of this group was 3, which occurred from June to October and means the office area was completely uncomfortable for the users in this period. Conversely, the minimum PMV was 1.61 in the office zone in the month of January, which, however, is still in the range of discomfort. As seen in Figure 7, the atrium zone of the southwest atrium orientation in the office building simulation model had its yearly PMV average with the 0% window-opening ratio as 1.11. The maximum PMV was 1.64 in October, and the minimum PMV was 0.58 in April. Additionally, the 25%, 50%, 75%, and 100% window-opening ratios had about the same PMV with an average value of 2.19, whereas the maximum PMV was 2.61 in October, and the minimum PMV was 1.25 in January.

As depicted in Figure 8, the office simulation building model of the southwest atrium orientation in the office zone with different window-opening ratios had yearly PPD averages as follows: 0% as 87.10% (PPD), 25% as 87.74% (PPD), 50% and 75% as 87.75% (PPD), and the 100% window-opening ratio as 90.21% (PPD). The common point of this group with all the different window-opening ratios occurred as 99.12% (PPD) from June to September. Conversely, the minimum PPD values were for 0% as 56.05% (PPD), 25% as 55.87% (PPD), 50% as 56% (PPD), and the 75% window-opening ratio as 56% (PPD), all in the month of January. Consequently, the office zone of this group may be stated to have a complete discomfort condition for the occupants. As shown in Figure 9, the PPD of the atrium zone in the southwest atrium orientation with different window-opening ratios was lower than the PPD of the office zone, although it was still characterized as discomfort. For instance, in this group, the 0% window-opening ratio had a PPD average of 37.98% for the year, whereas the maximum PPD was 55.86% in October, and the minimum PPD was 19.64% in April. Additionally, the 25%, 50%, 75%, and 100% window-opening ratios had similar approximate PPD values with a 74.12% yearly average, the maximum PPD of 87.33% in October, and the minimum PPD of 44.49% in January. Regardless, the atrium zone also had an uncomfortable space like in the office zone of this building simulation model group.

As illustrated in Figure 10, the simulation models of the northwest atrium orientation had a remarkable fluctuation in the monthly PMV values for different window-opening ratios. The office zone of this group with the 0% window-opening ratio had a 2.79 PMV average, the 3 PMV maximum which occurred from June to October, and the 1.97 PMV minimum in January. The 25% window-opening ratio in this group had the same yearly PMV average with the 100% window-opening ratio at 1.33 PMV. The maximum PMV for the 25% window-opening ratio was 2.98 in July, while the minimum PMV was 0.02 in January. When the 50% window-opening ratio was used, the office zone had a 1.27 PMV average during the year, the 2.95 maximum PMV in July, and the 0.14 minimum PMV in December. Furthermore, when the window-opening ratio was increased to 75%, the office space had its yearly average as 1.30, the maximum PMV as 2.92 in July, and the minimum PMV as 0.23 in April. As it may be seen in Figure 11, the atrium zone in the office building simulation model with the northwest atrium orientation and the 0% window-opening ratio had a yearly average of 2.61 PMV, the 2.82 PMV maximum in October, and the minimum of 2.17 PMV in January. Additionally,

the 25%, 50%, 75%, and 100% window-opening ratios had PMV averages of 0.93, 0.93, 1.04, and 1.11, respectively. The maximum PMV for the 25% window-opening ratio was 1.67, which was 1.6 for the 50% ratio, 1.62 for the 75% ratio, and 1.64 for the 100% ratio, all of which occurred in October. Conversely, the minimum PMV values at the 25% and 50% window-opening ratios were 0.07 and 0.22, respectively, both in January. The 75% and 100% window-opening ratios had their minimum PMV values as 0.49 and 0.58, respectively, both in April.

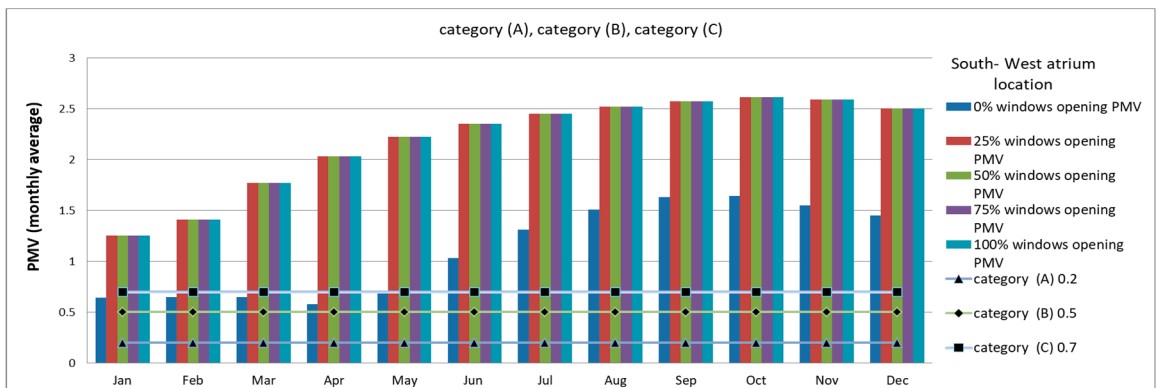

**Figure 6.** Predicted mean vote (PMV) for southwest atrium orientation of office space with categories A, B, and C monthly averages of metabolic rate (M) of 1.2 met, air speed of 0.15–0.3 m/s, and clothing value with 0.6–0.95 clo.

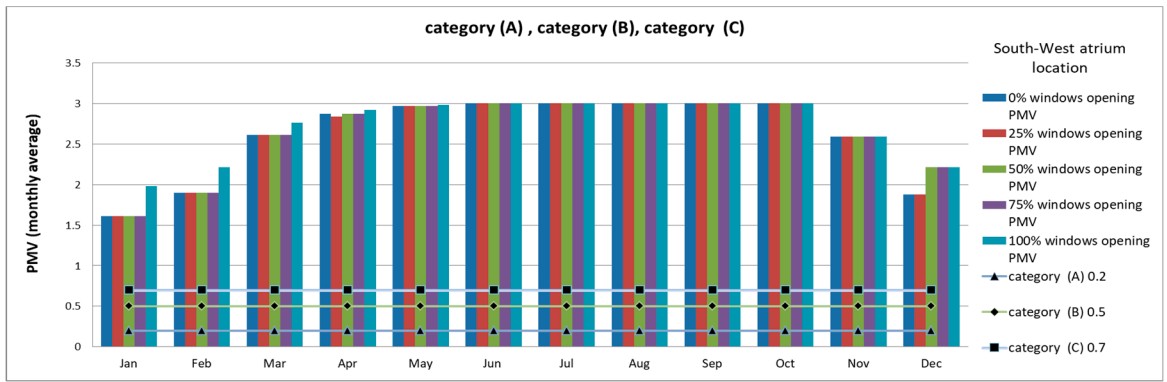

**Figure 7.** PMV for southwest atrium orientation of atrium space with categories A, B, and C monthly averages of M of 1.2 met, air speed of 0.15–0.3 m/s, and clothing value with 0.6–0.95 clo.

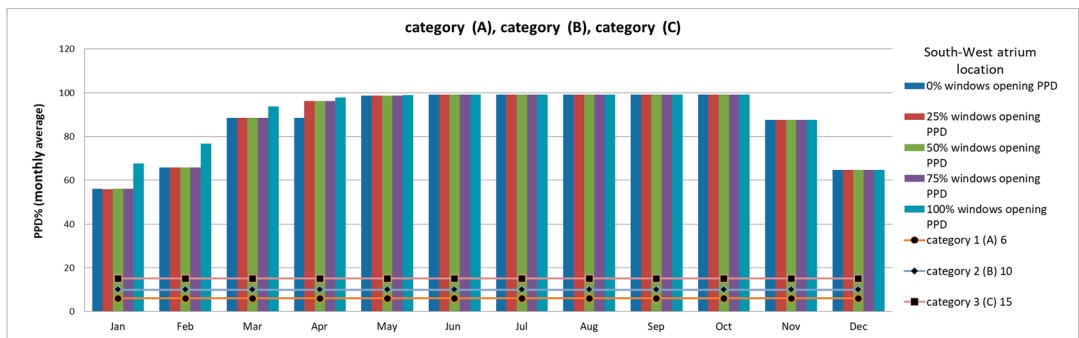

**Figure 8.** Predicted percentage of dissatisfied (PPD) for southwest atrium orientation of office space with categories A, B, and C monthly averages of M of 1.2 met, air speed of 0.15–0.3 m/s, and clothing value with 0.6–0.95 clo.

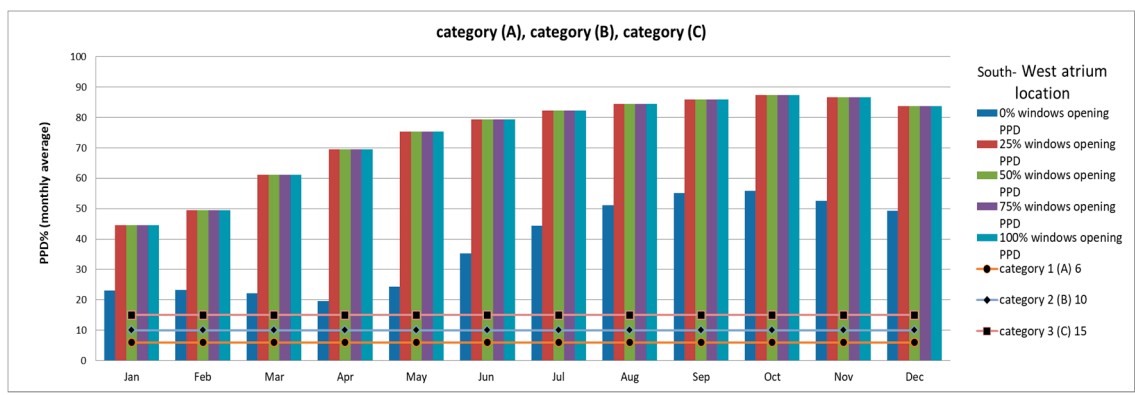

**Figure 9.** PPD for southwest atrium orientation of atrium space with categories A, B, and C monthly averages of M of 1.2 met, air speed of 0.15–0.3 m/s, and clothing value with 0.6–0.95 clo.

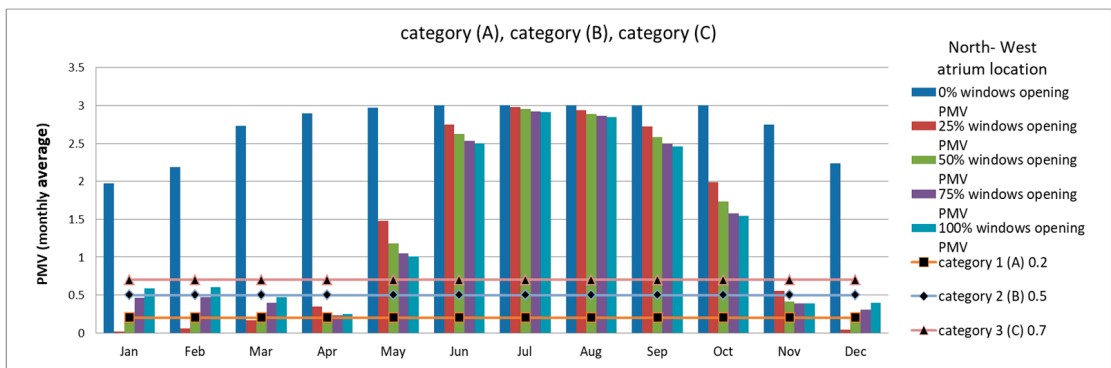

**Figure 10.** PMV for the northwest atrium location of office space with categories A, B, and C monthly averages of M of 1.2 met, air speed of 0.15–0.3 m/s, and clothing value with 0.6–0.95 clo.

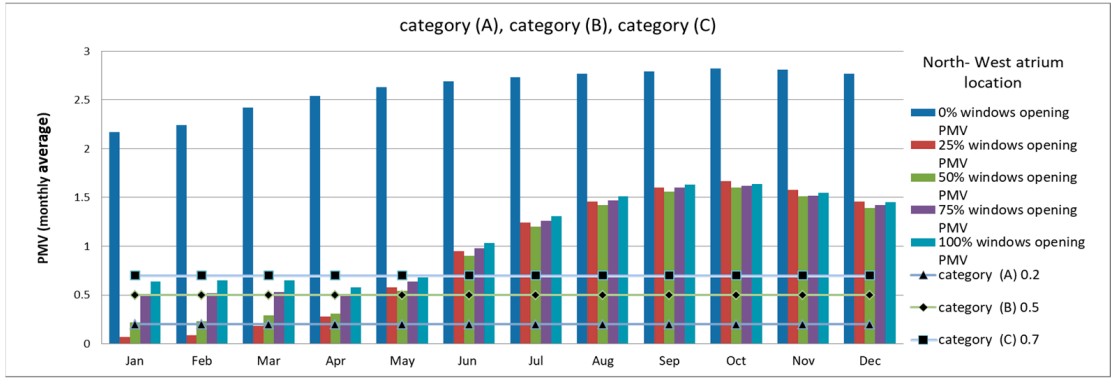

**Figure 11.** PMV for the northwest atrium location of atrium space with categories A, B and C monthly averages of M of 1.2 met, air speed of 0.15–0.3 m/s, and clothing value with 0.6–0.95 clo.

According to Figure 12, the office building simulation model with the northwest atrium orientation had the highest level of discomfort when all of the windows were completely closed (0% opening) in the office space, which had a 91.33% average PPD annually, 99.12% maximum PPD from June to October, and 66.91% minimum PPD in January. Additionally, the 25% window-opening ratio in the office zone had a 47.56% annual PPD average, the 98.79% maximum PPD in July which corresponded to complete discomfort, and the 7.5% minimum PPD in February in the comfort range. The 50%, 75%, and 100% window-opening ratios had the PPD parameter results of 45.07%, 45.68%, and 46.33%, respectively, for their yearly averages, and 98.34%, 97.95%, and 97.78% maximum PPD, respectively, all in the month of July. However, the minimum PPD for the 50% window-opening ratio was 8.52% in December, 9.8% in April for the 75% window-opening ratio, and 9.92% for the 100% window-opening

ratio, also in April. As depicted in Figure 13, the northwest atrium orientation in the office building simulation model with the 0% window-opening ratio had a complete discomfort condition in the atrium zone during the year with a 87.3% average PPD, the 93.6% maximum PPD in October, and the 72.5% minimum PPD in January. Furthermore, the 25%, 50%, 75%, and 100% window-opening ratios also had similar PPD results in the atrium zone. The 25% window-opening ratio in the atrium zone had a 33.9% yearly PPD average, 57.2% maximum PPD in October, and 8.3% minimum PPD in February. The 50% window-opening ratio in the atrium zone had a 32.6% yearly PPD average, 54.6% maximum PPD in October, and 10.2% minimum PPD in February. The 75% and 100% window-opening ratios in the atrium zone had 36.11% and 37.9% yearly PPD averages, 55.24% and 55.89% maximum PPD values in October, and 16.96% and 19.64% minimum PPD values, respectively, in April. As shown in Figure 14, the northeast atrium orientation of the simulation office building model with the 0% window-opening ratio in the office zone had a 2.73 yearly PMV average, 3 maximum PMV from June to October, and 1.98 minimum PMV in January. Furthermore, the office zone in this group with 25% and 50% window-opening ratios had yearly PMV averages of 1.32 and 1.26, 2.97 and 2.94 maximum PMV values both in the month of July, and 0.01 and 0.16 minimum PMV values in January and December, respectively. Additionally, the 75% and 100% window-opening ratios had yearly PMV average values of 1.92 and 1.33, maximum PMV values of 2.91 and 2.9 both in July, and minimum PMV values of 0.27 and 0.29 both in April. As it may be seen in Figure 15, the PMV of the building simulation model with the northeast atrium orientation when the windows were closed (0% opening) had a yearly average of 2.59, the maximum PMV of 2.8 in October and November, and the 2.15 minimum PMV in January, indicating complete discomfort for the users in the atrium zone. In contrast, the 25%, 50%, 75%, and 100% window-opening ratios in this simulation group had yearly PMV averages of 0.88, 0.92, 1.04, and 1.18, respectively, in the atrium space. Furthermore, the maximum and minimum PMV values for the 25% window-opening ratio were 1.6 maximum PMV in September and 0.05 minimum PMV in January; the 50% window-opening ratio had a 1.54 maximum PMV in October and a 0.26 minimum PMV in January; the 75% window-opening ratio had a 1.58 maximum PMV in October and a 0.52 minimum PMV in April; and the 100% window-opening ratio had a 1.63 maximum PMV in September and October and a 0.71 minimum PMV in April.

As shown in Figure 16, like with other office building simulation model groups, the office zone in the northeast atrium orientation simulation of the office building with the 0% window-opening ratio had a discomfort condition for the users during the year with a 91.3% yearly PPD average, 99.1% maximum PPD from June to October, and 67.1% minimum PPD in January. The 25% and 50% window-opening ratios had close PPD results, which were as follows: 47.1% yearly PPD average, 98.7 maximum PPD in July, and 7.7% minimum PPD in December for the 25% window-opening ratio, while the 50% window-opening ratio had a 44.73% yearly PPD average, 98.17% maximum PPD in July, and 9.37% minimum PPD in December. Additionally, the 75% and 100% window-opening ratios had 45.2% and 46.5% yearly PPD averages, maximum PPD values of 97.8% and 97.5% in July, and minimum PPD values of 10.7% and 11.7%, respectively, both of which occurred in April.

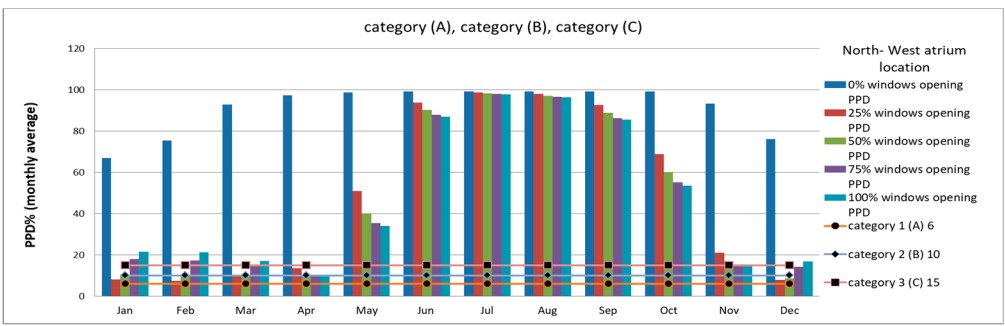

**Figure 12.** PPD for the northwest atrium location of office space with categories A, B, and C monthly averages of M of 1.2 met, air speed of 0.15–0.3 m/s, and clothing value with 0.6–0.95 clo.

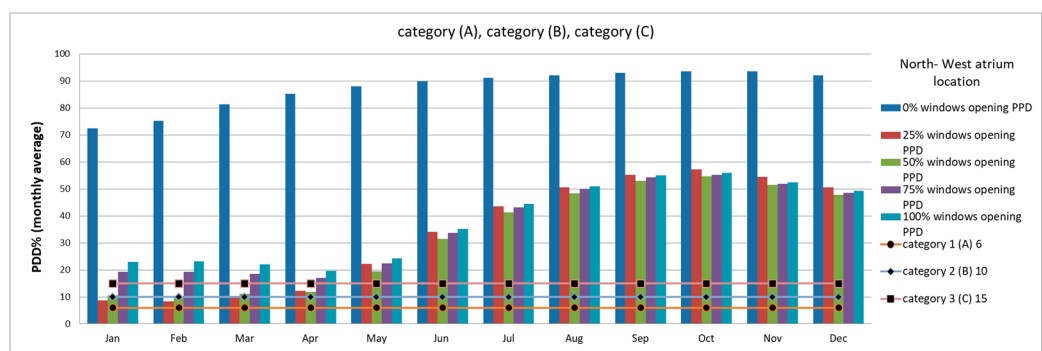

**Figure 13.** PPD for the northwest atrium location of atrium space with categories A, B, and C monthly averages of M of 1.2 met, air speed of 0.15–0.3 m/s, and clothing value with 0.6–0.95 clo.

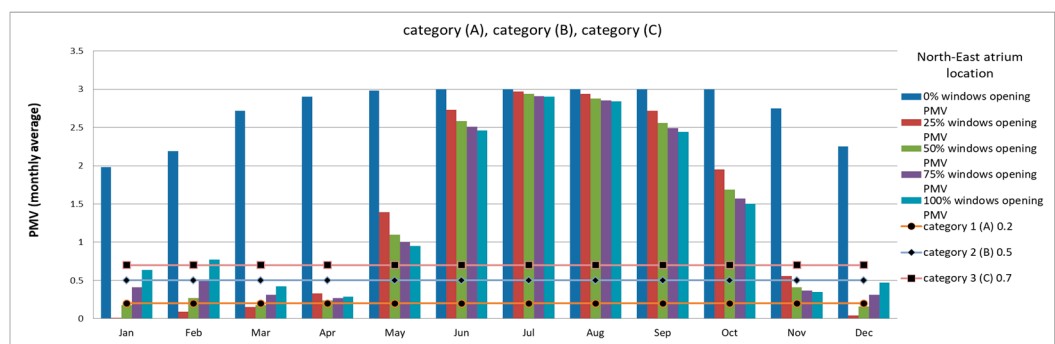

**Figure 14.** PMV for the northeast atrium location of office space with categories A, B, and C monthly averages of M of 1.2 met, air speed of 0.15–0.3 m/s, and clothing value with 0.6–0.95 clo.

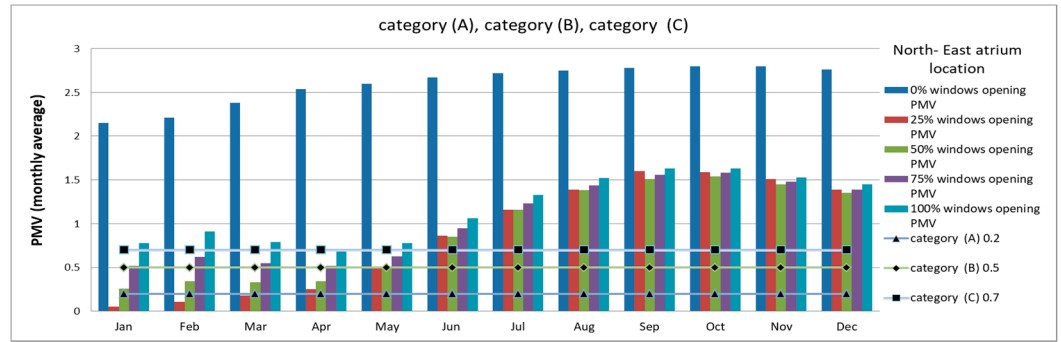

**Figure 15.** PMV for the northeast atrium location of atrium space with categories A, B, and C monthly averages of M of 1.2 met, air speed of 0.15–0.3 m/s, and clothing value with 0.6–0.95 clo.

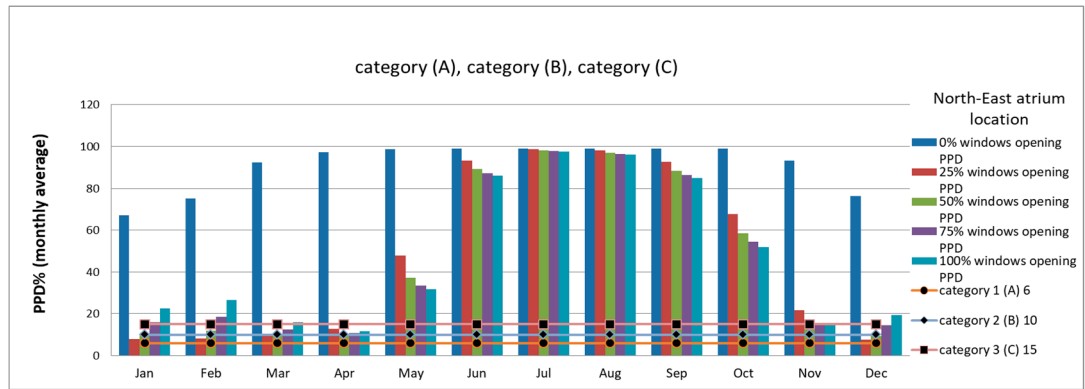

**Figure 16.** PPD for the northeast atrium location of office space with categories A, B, and C monthly averages of M of 1.2 met, air speed of 0.15–0.3 m/s, and clothing value with 0.6–0.95 clo.

As can be seen in Figure 17, both the atrium zone and the office zone (Figure 16) in the northeast atrium orientation of the office building simulation model had similar discomfort conditions when the windows were completely closed (0% opening) in both warm and cold seasons. The atrium zone had an 86.6% yearly PPD average, 93.2% maximum PPD in October, and 71.9% minimum PPD in January. The 25%, 50%, 75%, and 100% window-opening ratios had 32.36%, 32.06%, 35.3%, and 39.5% yearly PPD averages, and 54.8%, 52.7%, 53.5%, and 55.3% maximum PPD respectively, which all occurred in October, and minimum PPD values of 8.2% in January, 11.2% in January, 17.4% in April, and 23.4% in April, respectively. As illustrated in Figure 18, the southeast atrium orientation of the office building simulation had fluctuations in the thermal comfort of the office zone. The 0%, 25%, and 50% window-opening ratios in the office zone for this group had 2.5, 1.2, and 1.2 yearly PMV averages, respectively.

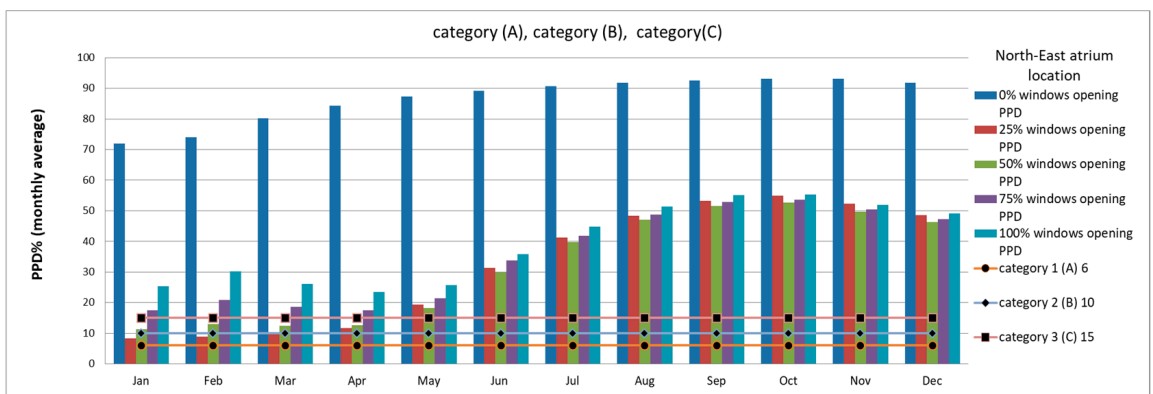

**Figure 17.** PPD for the northeast atrium location of atrium space with categories A, B, and C monthly averages of M of 1.2 met, air speed of 0.15–0.3 m/s, and clothing value with 0.6–0.95 clo.

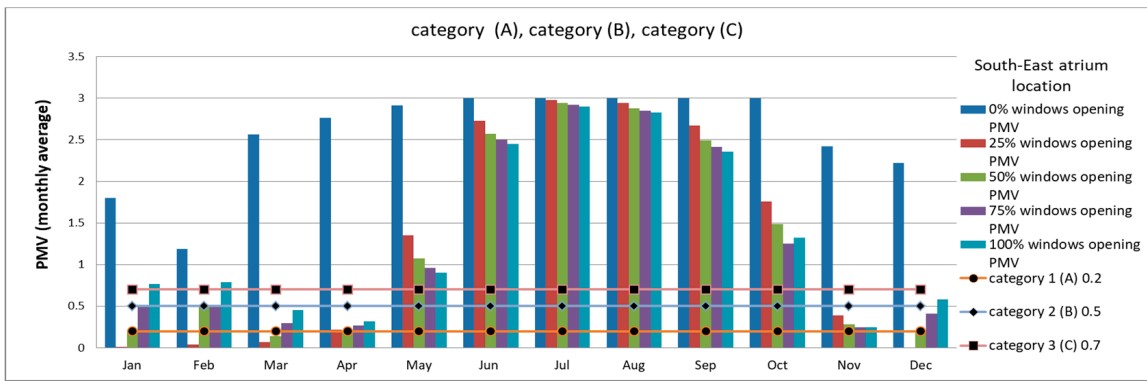

**Figure 18.** PMV for the southeast atrium orientation of office space with categories A, B, and C monthly averages of M of 1.2 met, air speed of 0.15–0.3 m/s, and clothing value with 0.6–0.95 clo.

Additionally, the maximum PMV was 3 for the 0% window-opening ratio from June to October, 2.9 for the 25% window-opening ratio in July, and 2.9 for the 50% window-opening ratio in July. Moreover, the minimum PMV was 1.1 in February (0% opening), 0 in December (25% opening), and 0.1 in March (50% opening). The 75% and 100% window-opening ratios had similar PMV values with 1.2 and 1.3 yearly PMV averages, and the 2.9 and 2.9 maximum respective PMV values, both in July. Additionally, the minimum PMV values for both the 75% and 100% window-opening ratios were 0.2 in November. As it may be seen in Figure 19, the southeast atrium orientation of the office building simulation model in the atrium zone with the 0% window-opening ratio had a 2.1 yearly average, 2.5 maximum PMV in October, and 1.1 minimum PMV in January; the 25% window-opening ratio had a 0.7 yearly PMV average, 1.4 maximum PMV in October, and 0.04 minimum PMV in January; the 50% window-opening ratio had a 0.9 yearly PMV average, 1.4 maximum PMV in September, and 0.3

minimum PMV in April; the 75% window-opening ratio had a yearly PMV average of 1, 1.5 maximum PMV in September, and 0.5 minimum PMV in April; and the 100% window-opening ratio had a 1.2 yearly PMV average, 1.6 maximum PMV in September, and 0.7 minimum PMV in April.

As depicted in Figure 20, the office zone of the southeast atrium orientation of the office building simulation with the 0% window-opening ratio had a 91.3% yearly PPD average, 99.1% maximum PPD from June to October, and 66.9% minimum PPD in January; the 25% window-opening ratio had a 47.5% yearly PPD average, 98.7% maximum PPD in July, and 7.5% minimum PPD in February; the 50% window-opening ratio had a 45% yearly PPD average, 98.3% maximum PPD in July, and 8.5% minimum PPD in December; the 75% window-opening ratio had a 45.6% yearly PPD average, 97.9% maximum PPD in July, and 9.8% minimum PPD in April; and the 100% window-opening ratio had a 46.3% yearly PPD average, 97.7% maximum PPD in July, and 9.9% minimum PPD in April. As shown in Figure 21, the PPD in the atrium zone of the southeast atrium orientation in the office building simulation models with the 0% window-opening ratio had a 72.8% yearly PPD average, 86.7% maximum PPD in October, and 40.3% minimum PPD in January; the 25% window-opening ratio had a 30.4% yearly PPD average, 51.9% maximum PPD in October, and 8.5% minimum PPD in March; the 50% window-opening ratio had a 33.3% yearly average PPD, 51.8% maximum PPD in September, and 14.4% minimum PPD in April; the 75% window-opening ratio had a 37.7% yearly average PPD, 53.9% maximum PPD in September, and 21.3% minimum PPD in April; and the 100% window-opening ratio had a 56.2% yearly PPD average, 56.2% maximum PPD in September, and 28% minimum PPD in April.

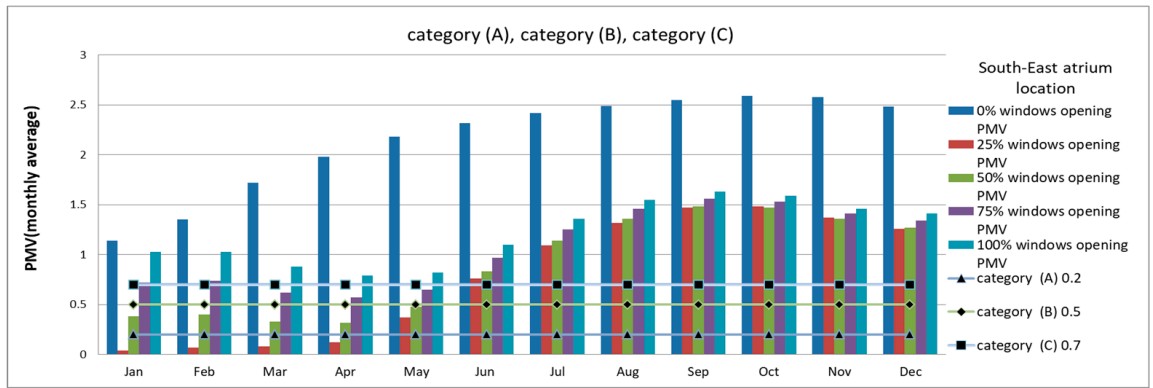

**Figure 19.** PMV for the southeast atrium orientation of atrium space with categories A, B, and C monthly averages of M of 1.2 met, air speed of 0.15–0.3 m/s, and clothing value with 0.6–0.95 clo.

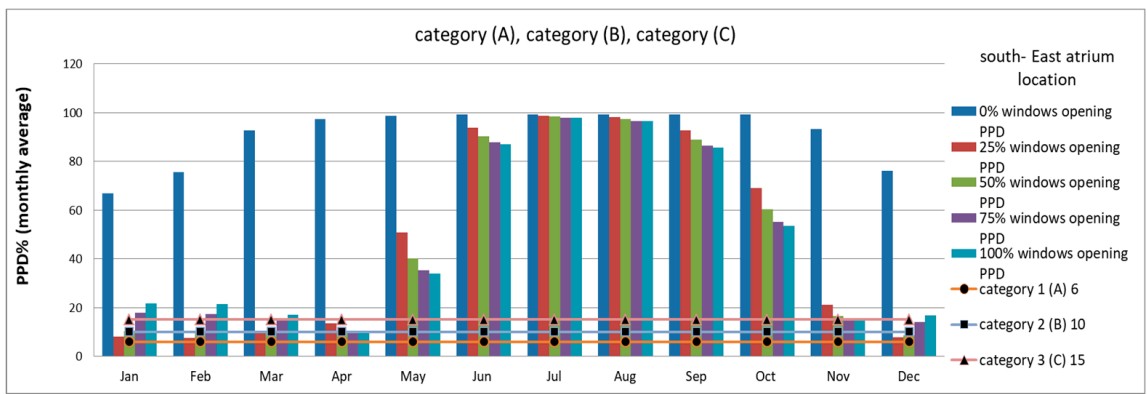

**Figure 20.** PPD for the southeast atrium orientation of office space with categories A, B, and C monthly averages of M of 1.2 met, air speed of 0.15–0.3 m/s, and clothing value with 0.6–0.95 clo.

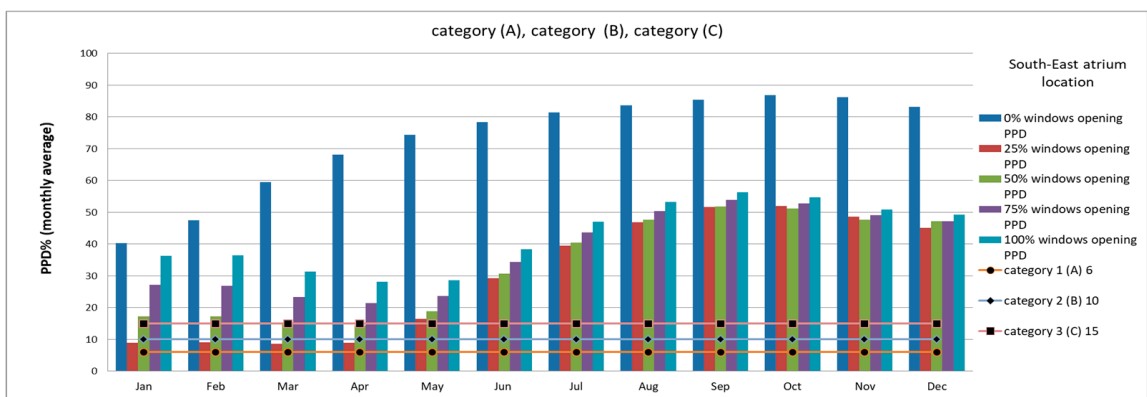

**Figure 21.** PPD for the southeast atrium orientation of atrium space with categories A, B, and C monthly averages of M of 1.2 met, air speed of 0.15–0.3 m/s, and clothing value with 0.6–0.95 clo.

As it may be seen in Tables 1 and 2, the northwest atrium orientation of the office building simulation model had a suitable office temperature with the 0% window-opening ratio at 24.2 °C when the external temperature was 10.9 °C, although the indoor temperature decreased slightly based on the increase in the window-opening ratio. Moreover, the office space's relative humidity was 41.7% with the 0% opening ratio and increased with the window-opening ratio in the same month by up to 53.5%. The 25% window-opening ratio with the same parameters had its internal temperature as 15.1 °C, PMV as 0.02, and PPD as 8% in January. As such, the office building simulation model with the northwest atrium in winter time appeared to have functioned better than other model simulations. Furthermore, the 50% window-opening ratio in the northwest atrium orientation had 10% < PPD < 10.35% and 0.17 < PMV < 0.21 in January, February, March, and April, and may, thus, be proposed as an acceptable window-opening ratio. The northwest atrium orientation of the office building simulation with the 50% window-opening ratio performed acceptably as a result of its thermal comfort in category A of PMV in January, February, March, April, and December. Additionally, the 25% window-opening ratio had a comfortable thermal condition based on PMV in category A throughout January, February, March, and December. The PMV of the 75% window-opening ratio had thermal comfort according to category B in January, February, March, April, November, and December. In contrast to the other window-opening ratios in the office building simulation, however, the northwest atrium orientation with the 0% window-opening ratio provided thermal comfort for the users in January, February, March, April, November, and December according to category C. Additionally, the northwest atrium orientation simulation model with the 25% window-opening ratio had better thermal comfort for users in the office and atrium zones in January, February, and March. It also had a lower minimum PMV (0.02) and PPD (8%) than other window-opening ratios, although the internal temperature and internal relative humidity fluctuated in the office space. Furthermore, in April, the dynamic simulation model with the 75% window-opening ratio had its PMV as 0.2 and PPD as 9.8% when the external temperature was 16.2 °C, the office space temperature as 17.5 °C and the atrium space temperature as 17.45 °C, and the relative humidity in office and atrium spaces as 63.4% and 64%, respectively. In contrast, the 0% and 25% window-opening ratios had the maximum dissatisfaction condition in May. This situation dramatically changed to the users' satisfaction throughout June, July, and August, while the 75% window-opening ratio simulation model with the same atrium orientation had a better thermal comfort condition in September and October.

**Table 1.** The predicted percentage of dissatisfied (PPD) values of the office building simulations during a year, which achieved thermal comfort according to categories A, B, and C.

| Atrium Orientation | Window-Opening Ratio | Months | PPD (%) Category |
|---|---|---|---|
| Northwest atrium (office space) | | Jan, Feb, Dec | (A) <6 |
| | 25% | Jan, Feb, March, Dec | (B) <10 |
| | | Jan, Feb, March, April, Dec | (C) <15 |
| | 50% | Jan, Feb, March, April, Dec | (B) <10 |
| | 50% | Jan, Feb, March, April, Nov, Dec | (C) <15 |
| | 75% | Feb, March, April, Nov, Dec | |
| | 75%, 100% | April | (B) <10 |
| | 100% | March, April, Nov, Dec | (C) <15 |
| Northwest atrium (atrium space) | 25%, 50% | Jan, Feb, March, April | (B) <10 |
| | 75% | April | (C) <15 |
| Northeast atrium (office space) | 25% | Jan, Feb, Dec | (A) <6 |
| | 25%, 50% | Jan, Feb, March, April, Dec | (B) <10 |
| | 50%, 75% | Jan, Feb, March, April, Nov, Dec | |
| | 25%, 50%, 75% | Jan, Feb, March, April, Dec | (C) <15 |
| | 100% | March, April, Nov, Dec | |
| Northeast atrium (atrium space) | 25% | Jan, Feb, March | (B) <10 |
| | 25%, 50% | Jan, Feb, March | |
| Southeast atrium (office space) | 25%, 50% | Jan, Feb, March, April | (C) <15 |
| | | Jan, Feb, Dec | (A) <6 |
| | 25% | Jan, Feb, March, Dec | (B) <10 |
| | 50% | Jan, Feb, March, April, Nov, Dec | |
| | 75%, 100% | April | |
| Southeast atrium (atrium space) | 25% | Jan, Feb, March, April, Dec | (C) <15 |
| | 50% | Jan, Feb, March, April, Nov, Dec | |
| | 75%, 100% | March, April, Nov, Dec | |
| | 25% | Jan, Feb, March, April | (B) <10 |
| | 25%, 50% | Jan, Feb, March, April, May | (C) <15 |

**Table 2.** The predicted mean vote (PMV) values of the office building simulations during a year, which achieved thermal comfort according to categories A, B, and C.

| Atrium Orientation | Window-Opening Ratio | Months | PMV Category |
|---|---|---|---|
| Northwest (office space) | | Jan, Feb, March, Dec | (A) −0.20 to 0.20 |
| | 25% | Jan, Feb, March, April, Nov | |
| | | Jan, Feb, March, April, Nov, Dec | (C) −0.70 to 0.70 |
| | 50% | Jan, Feb, March, April, Dec | (A) −0.20 to 0.20 |
| | 50%, 75%, 100% | Jan, Feb, March, April, Nov, Dec | (C) −0.70 to 0.70 |
| | 25%, 50% | Jan, Feb, March, April | (B) −0.50 to 0.50 |
| Northwest (atrium space) | 25% | Jan, Feb, March | (A) −0.20 to 0.20 |
| | 50% | Jan, Feb | |
| | 75% | Jan, Feb, March, April | (B) −0.50 to 0.50 |
| | 100% | Jan, Feb, March, April, May | (C) −0.70 to 0.70 |
| Northeast (office space) | 25% | Jan, Feb, March, Dec | (A) −0.20 to 0.20 |
| | 50% | Jan, March, April, Dec | |
| | 25% | Jan, Feb, March, April, Dec | (B) −0.50 to 0.50 |
| | 50% | Jan, Feb, March, April, Nov, Dec | |
| | 25%, 50%, 75%, 100% | Jan, March, April, Nov, Dec | (C) −0.70 to 0.70 |
| Northeast (atrium space) | 25% | Jan, Feb, March | (B) −0.50 to 0.50 |
| | 25%, 50% | Jan, Feb, March, April, May | (C) −0.70 to 0.70 |
| | 75% | | |
| Southeast (office space) | 25% | Jan, Feb, March, April, Dec | (A) −0.20 to 0.20 |
| | 50% | March, April, Dec | |
| | 25%, 50% | | (B) −0.50 to 0.50 |
| | 75% | Jan, Feb, March, April, Nov, Dec | |
| | 25%, 50%, 75% | | |
| | 100% | March, April, Nov, Dec | (C) −0.70 to 0.70 |
| Southeast (atrium space) | 25% | Jan, Feb, March, April | (A) −0.20 to 0.20 |
| | 25%, 50% | Jan, Feb, March, April, May | (B) −0.50 to 0.50 |
| | 25%, 50%, 75% | | (C) −0.70 to 0.70 |

The 0% window-opening ratio of the northeast atrium orientation in the office building simulation model had a negative thermal condition, especially in January, February, and March. Furthermore, this simulation model with the 50% and 75% window-opening ratios had a better function in April for which the indoor temperature was 17.8 °C. Then, the same results were obtained in May, June, July, and August when the indoor temperature, indoor relative humidity, PMV, and PPD dramatically increased. Meanwhile, the 75% and 25% window-opening ratios had a more acceptable thermal comfort level than other dynamic simulation model groups in October and November, and in December, respectively. According to the categories, the office building simulation model with the southeast atrium orientation and 50% window-opening ratio had a lower PMV than other window-opening ratios with category A results throughout January, March, April, November, and December. The 75% window-opening ratio had a category B PMV in January, February, March, April, November, and December, indicating better thermal comfort. Overall, the 25%, 50%, 75%, and 100% window-opening ratios had an even better thermal condition based on their PMV results in category C in January, February, March, April, November, and December. For the office building simulation model with the northeast atrium orientation, the PMV of the 50% window-opening ratio was category A in January, February, March, April, and December, indicating thermal comfort. The thermal comfort condition of the 75% window-opening ratio was based on category B throughout January, February, March, April, November, and December. Furthermore, the PMV of the 100% window-opening ratio with the northeast atrium orientation had acceptable comfort in January, February, March, April, November, and December based on category C. The office building simulation models with the southeast atrium orientation and 100% window-opening ratio had the highest degree of dissatisfaction (67% PPD and 1.98 PMV) in the office space and atrium space in comparison to other simulation models with the same atrium orientation but with different ratios of window opening. The model with the 100% window-opening had an office space temperature of 24.8 °C and atrium space temperature for 17.5 °C when external temperature was 10.2 °C. Furthermore, the office space's relative humidity was 42.2%. The simulation models of the 0%, 25%, 50%, and 75% window-opening ratios had a PMV of approximately 1.61, while the 25% window-opening ratio model with the same atrium orientation had the lowest PPD (55.8%) relative to those in the other ratios. Throughout February, the users' dissatisfaction increased slightly in both PMV (1.90) and PPD (65.8%). Additionally, the atrium space of the southeast simulation model had an acceptable condition for PMV and PPD in January, February, March, and April based on the categories. The southeast atrium orientation of the simulation models with the 25% and 50% window-opening ratios had PMV thermal comfort levels based on category A in January, February, March, April, and December. These simulation models also had occupants' comfort condition based on category B in November. While the PMV of the 75% window-opening ratio was in category B for January, February, March, April, November, and December, the 100% window-opening ratio was in category C in January, February, March, November, and December, indicating complete thermal comfort. The PPD values of the 25% and 50% window-opening ratios in January, February March, April, and November had thermal comfort based on category A. Moreover, the comfort zones of the 75% and 100% window-opening ratios were based on category C in March, April, November, and December. Additionally, while the office space temperature increased dramatically, the relative humidity remained quite the same. In March, April, and May, the PPD value increased greatly especially in the simulation model using the 100% window-opening ratio. Furthermore, in June, July, and August, the maximum total of the PMV was 3 and PPD was 99.1%, but the relative humidity in the office space decreased despite a high indoor temperature, particularly for the 0% window-opening ratio simulation.

As seen in Table 3, the negative average of percentage of dissatisfied (PD) due to draught (%) in all the atrium building simulation models happened when all the windows were completely closed (0% opening). For instance, the northwest atrium orientation in the atrium space had the maximum user dissatisfaction result, especially in October and November, although this atrium orientation with the 25% window-opening ratio had the highest user satisfaction in January and February according to

the categories. This trend continued as the office space with the 0% window-opening ratio had the maximum dissatisfaction (PD) due to draught from April to November, but the highest average of satisfaction with the 25% window-opening ratio was in March and December. The building simulation model with the northwest atrium orientation had the average yearly dissatisfied (PD) due to draught (%) in the atrium zone in October, November, and December, and the office zone had its worst interior comfort levels from May to November (0% window-opening ratio). Conversely, the 50% window-opening ratio in this simulation group had the comfort condition in both atrium and office zones from June to May and December. Furthermore, the simulation model with the southeast atrium orientation had the same negative internal comfort with the 0% window opening ratio in both zones from October to November (atrium zone) and June to October (office zone). However, the occupants' maximum internal comfort satisfaction levels occurred with the 25% window-opening ratio in the atrium space in March and April, and the 50% window-opening ratio in the office space in December. However, these aforementioned results dramatically changed in the building simulation with the southwest atrium orientation. For example, in the atrium zone, the average comfort levels of the 25%, 50%, 75%, and 100% window-opening ratios occurred in October, while that for the 50% and 75% window-opening ratios occurred from June to September in the office zone. Additionally, the 0% window-opening ratio had thermal comfort in both zones (atrium and office spaces) in April and January in comparison to other simulation models.

**Table 3.** The percentage of dissatisfied (PD) due to draught (%) for the office building simulations during a year, which determined thermal comfort according to categories A, B, and C.

| Atrium Orientation | Window-Opening Ratio | Months | PDD Draught (%) Category |
|---|---|---|---|
| Northeast (office space) | 25% | Jan, Feb, March, Dec | (A) <10 |
| | | April | (B) <20 |
| | | Nov | (C) <30 |
| | 50% | Jan, Feb, April, Nov | (B) <20 |
| | | March, Dec | (A) <10 |
| | 75% | Jan, Feb, March, April, Nov, Dec | (B) <20 |
| | 100% | Jan, Feb | (C) <30 |
| | | March, April, Nov, Dec | (B) <20 |
| Northeast (atrium space) | 25% | Jan, Feb | (A) <10 |
| | | March, April, May | (B) <20 |
| | 50% | Jan, Feb, March, April, May | |
| | | Jun | (C) <30 |
| | 75% | Jan, March, April | (B) <20 |
| | | Feb, May | (C) <30 |
| | 100% | Jan, March, April, May | |
| Northwest (office space) | 25% | Jan, Feb, March, April, Dec | (A) <10 |
| | 50% | Dec | |
| | 75% | April | |
| | 25% | | |
| | 50% | Jan, Feb, March, Nov, Dec | (B) <20 |
| | 75% | | |
| | 100% | March, Nov, Dec | |
| Northwest (atrium space) | 25% | Jan, Feb, March | (A) <10 |
| | | April | (B) <20 |
| | | May | (C) <30 |
| | 50% | Jan, Feb, March, April, May | |
| | 75% | Jan, Feb, March, April | (B) <20 |
| | 100% | April | |
| | 75% | May | |
| | 100% | Jan, Feb, March, May | (C) <30 |
| | 100% | Jan, Feb | |

**Table 3.** *Cont.*

| Atrium Orientation | Window-Opening Ratio | Months | PDD Draught (%) Category |
|---|---|---|---|
| Southeast (office space) | 25% | Jan, Feb, March, Dec | (A) <10 |
| | 50% | Dec | |
| | 75% | April | |
| | 100% | April | |
| | 25% | April | (B) <20 |
| | 50% | Jan, Feb, March, April, Nov | |
| | 75% | Jan, Feb, March, Nov, Dec | |
| | 100% | March, Nov, Dec | |
| | 25% | Nov | (C) <30 |
| | 100% | Jan, Feb | |
| Southeast (atrium space) | 25% | Jan, Feb, March, April | (A) <10 |
| | 25% | May | (B) <20 |
| | 25% | Jun | (C) <30 |
| | 50% | Jan, Feb, March, April, May | |
| | 75% | Jan, Feb, March, April, May | (B) <20 |
| | 100% | April, May | |
| Southwest (atrium space) | 0% | Jan, Feb, March, April, May | (C) <30 |

As illustrated in Table 4, the dynamic simulation model with the northwest atrium orientation had a better user comfort level during the year than other simulation model groups. The northwest, northeast, and southeast atrium orientations with the 0% window-opening ratio had the most negative thermal comfort conditions in the atrium and office spaces, especially in October, and from June to October. Furthermore, the 50% window-opening ratio with the northwest atrium orientation had a comfort condition in February, while the northeast and southeast atrium orientations had internal comfort in December. As depicted in Table 5, the 0% window-opening ratio in all simulation models had the maximum average yearly dissatisfaction in the PD levels due to cool or warm floor (%), especially in the atrium zone of the northeast, northwest, and southeast atrium orientation models in October and November. Additionally, the office zone with the same parameters as in the previous group had its maximum dissatisfaction levels of PD due to cool or warm floor (%) in the northeast, northwest, and southeast atrium orientation models from June to October. In contrast, the atrium zone of the 50% window-opening ratio simulation model of the northeast atrium orientation (from January to April) and northwest atrium orientation (January and February), and the 25% window-opening ratio of the southeast atrium orientation (March and April) had the most acceptable thermal comfort levels. Additionally, most users' satisfaction in the office zone occurred with the 50% window-opening ratio in the northeast atrium orientation from January to April and in December, and, in the northwest and southeast atrium orientation condition, this happened in December for both the window-opening ratios. As can be seen in Table 6, the PD due to radiant temperature asymmetry (%) in the office building simulation for all the groups was the most negative in terms of the yearly and monthly average of users' dissatisfaction for the 0% window-opening ratio in the northeast, southeast, and northwest atrium orientations in the atrium zone in October and November, and in the office zone from June to October. Despite this, the 50% window-opening ratio had the maximum user satisfaction average levels in both zones from January to April.

**Table 4.** The PD due to vertical air temperature difference (%) of the office building simulations during a year, which indicated thermal comfort according to categories A, B, and C.

| Atrium Orientation | Window-Opening Ratio | Months | PDD Vertical Air Temperature Difference (%) |
|---|---|---|---|
| Northwest (office space) | 25%<br>75%<br>100% | Jan, Feb, March, Dec<br>April | |
| Northwest (atrium space) | 25% | Jan, Feb, March<br>Jan, Feb, March, Dec | |
| Northeast (office space) | 50% | March, Dec | (C) <10 |
| Northeast (atrium space) | 25% | Jan, Feb<br>Jan, Feb, March, Dec | |
| Southeast (office space) | 50%<br>75%<br>100% | Dec<br>April | |
| Southeast (atrium space) | 25% | Jan, Feb, March, April | |

**Table 5.** The PD due to cool or warm floor (%) of the office building simulations during a year, which provided thermal comfort according to categories A, B, and C.

| Atrium Orientation | Window-Opening Ratio | Months | PDD Cool or Warm Floor (%) |
|---|---|---|---|
| Northeast (office space) | 25%<br>50%<br>25%<br>50%<br>75%<br>100% | Jan, Feb, March, Dec<br>March, Dec<br>April<br>Jan, Feb, April<br>March, April, Dec<br>April, Nov | (A) <10 (B) <10<br><br>(C) <15 |
| Northeast (atrium space) | 25%<br>50% | Jan, Feb<br>March, April<br>Jan, Feb, March, April | (A) <10 (B) <10<br>(C) <15 |
| Northwest (office space) | 25%<br>50%<br>75%<br>100%<br>25%<br>50% | Jan, Feb, March, Dec<br>Dec<br><br>April<br><br>Jan, Feb, March, April | (A) <10 (B) <10<br><br>(C) <15 |
| Northwest (atrium space) | 25%<br>50% | Jan, Feb, March<br>April<br>Jan, Feb, March, April | (A) <10 (B) <10<br>(C) <15 |
| Southeast (office space) | 25%<br>50%<br>75%<br>100%<br>25%<br>50%<br>75%<br>100% | Jan, Feb, March, Dec<br>Dec<br><br>April<br><br>Jan, Feb, March, April<br>March, Dec<br>Nov | (A) <10 (B) <10<br><br><br>(C) <15 |

**Table 6.** The PD due to radiant temperature asymmetry (%) of the office building simulations during a year, which provided thermal comfort according to categories A, B, and C.

| Atrium Orientation | Window-Opening Ratio | Months | PDD Radiant Temperature Asymmetry (%) |
|---|---|---|---|
| Northeast (office space) | 25% <br> 50% | Jan, Feb, March, Dec <br> March, Dec | |
| Northeast (atrium space) | 25% | Jan, Feb <br> Jan, Feb, March | |
| Northwest (office space) | 75% <br> 100% | April | (C) <10 |
| Northwest (atrium space) | 25% | Jan, Feb, March <br> Jan, Feb, March, Dec | |
| Southeast (office space) | 50% <br> 75% <br> 100% | Dec <br> April | |
| Southeast (atrium space) | 25% | Jan, Feb, March, April | |

## 7. Conclusions

On a general note, this study discovered that the northeast atrium orientation in the office building simulation model had a sufficient energy performance and suitable thermal comfort for users throughout the year in comparison to other dynamic simulation models. Additionally, the window-opening ratio had a direct relationship with the indoor relative humidity; the more the window was open, the more dramatic the increase in the relative humidity was. On the other hand, because of the glass wall building, which consisted of two glass layers, the air in the middle part could not control the temperature, thus causing the indoor temperature to change slightly in different months. Overall, the atrium was both suitable and effective in terms of indoor thermal comfort as indicated by actual temperatures and the MRT (mean radiant temperature). The southeast atrium orientation in the simulation office models had a more suitable office space temperature than other simulation models in spite of the approximately 51% office indoor relative humidity in the cold season. In the summer, the southeast atrium orientation in the simulation office models had better thermal comfort levels than other atrium orientation models in terms of temperature. As an example, the average office zone temperatures in June with the 25%, 50%, 75%, and 100% window-opening ratios were 28.2 °C, 27.2 °C, 26.9 °C, and 26.7 °C, respectively, when the external temperature was 26 °C, although the office zone relative humidity increased to 62%.

The northeast atrium orientation of the simulation models and the 25% and 50% window-opening ratios had a thermal comfort condition throughout the cold season based on category C regarding their PPD values. Furthermore, during the cold season, the 25% window-opening ratio had 137.2 W heat loss and 189.5 W heat gain in the office zone, and 37.7 W heat loss and 204.7 W heat gain in atrium zone. Additionally, with the same atrium orientation in the same season, the 50% window-opening ratio had 134.5 W heat loss and 134.2 W heat gain in the office zone, and 40 W heat loss and 192 W heat gain in the atrium zone. The simulation models with the southeast atrium orientation and 25%, 50%, and 75% window-opening ratios had PMV values based on category C in January, February, March, and April (cold months), indicating the occupants' comfort in all zones. The energy performance of this atrium orientation group with the 25% window-opening ratio had 108.7 W heat loss and 143 W heat gain in the office zone, and 68.8 W heat loss and 130 W heat gain in the atrium zone. The 50% window-opening ratio had 111.7 W heat loss and 142.7 W heat gain in the office zone, and 67.5 W heat loss and 121.2 W heat gain in the atrium zone, while the 75% window-opening ratio had 137 W heat loss and 142.5 W heat gain in the office zone, and 66.7 W heat loss and 118.2 W heat gain in the atrium zone. Furthermore, the PPD of the office zone in this group of simulations was based on category C for thermal comfort in January, February, March, April, and December (cold months). These parameters were the same in the atrium zone, which had thermal comfort in the same months except

for December. Unfortunately, during the warm season, there were no simulation models that had a completely comfortable condition according to their PMV and PPD values, especially in the summer.

According to the energy performance parameters of building heat gain, loss, and mean radiant temperature, the dynamic simulation models of the northwest atrium orientation in the office building with 0% window-opening ratio had the worst yearly average energy performance with identical heat loss and gain values of 103.4 W, and also mean radiant temperatures of about 34.4 °C in the office and 36.7 °C in the atrium zone. However, the southeast atrium orientation and northeast atrium orientation in the office building with a 0% window-opening ratio had a negative energy performance behavior. With regard to thermal comfort parameters, it can be mentioned that the southwest atrium orientation of the office building simulation models in the office zone with 25%, 50%, 75%, and 100% window-opening ratios in September, October, November, and December, and in the atrium zone from May to October did not provide thermal comfort according to predicted mean vote. Furthermore, this simulation model group with 25%, 50%, 75%, and 100% window-opening ratios did not provide thermal comfort in the office zone from May to October and the atrium zone in September, October, and November based on predicted percentage dissatisfied.

Generally, the 0% window-opening ratio was significantly dissatisfying for the users' thermal comfort in terms of PMV and PPD based on the hot and humid climate. This means that, in this microclimate, it is vital to have air movement, although the 100% window-opening ratio could act as a good solution for the atrium areas in the buildings. Moreover, the 25% window-opening ratio with different atrium orientations in the office building performed better with regard to increasing the indoor quality by improving the indoor temperature and relative humidity. As an example, the northeast and southeast atrium orientations in the office building had better internal comfort features than other simulation models, while also having the minimum average values of PMV and PPD. It may be summarized that the 25% and 50% window-opening ratios had the highest user comfort condition results throughout the year. The window-opening ratio may be useful for generating thermal comfort by regulating air movement and decreasing humidity. The northwest atrium orientation in the simulation building with the 25%, 50%, and 75% window-opening ratios had thermal comfort conditions during the cold season according to PMV based on category C. Additionally, the 25% and 50% window-opening ratios for this group had PPD results based on category C, indicating the users' comfort in the entirety of the building throughout the cold season. The office building simulation models with the northeast atrium orientation and the 25%, 50%, 75%, and 100% window-opening ratios had thermal comfort in the office zone in January, February, March, and April (cold season) according to PMV based on category C. These comfort conditions continued in the atrium zone with all previous parameters except for the 100% window-opening ratio.

It is noteworthy that, by considering the architectural parameters in a single office atrium building (atrium placement and window-opening ratios), one can increase the users' comfort dramatically and improve energy efficiency performance throughout the year in Gazimagusa, North Cyprus. Also, applying other parameters like shading devices or blind systems can have more advantages, but it is essential to assess the net action of the atrium building behavior itself in this climate in moving toward sustainability.

**Author Contributions:** H.Z.A. and R.A. conceived and designed the study; R.A. performed the experiments and simulations, and wrote the paper; H.Z.A. supervised the study, and provided sources, comments, and major edits for the paper.

**Funding:** This research received no external funding.

**Conflicts of Interest:** The authors declare no conflict of interest.

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
