# Peer review of "Thermal Comfort and Energy Performance of Atrium in Mediterranean Climate"

_sustainability, doi:10.3390/su11041213_

Round 1
Reviewer 1 Report
I am very sorry to say the following frankly:
To me this is like a dissertation report, without new understanding brought to this field. Dynamic modelling now has been routinely applied to new building design in many countries due to widely accessibility of the programs like TAS.
There is no novelty in application and methodology.
The model needs validation before discuss its prediction results.
No impact to the research communit. Posible good for practitioners or architects and clients who need to be convinced that dynamic thermal modelling is essential to create a good design that provides comfort with less energy.

Author Response
Reviewer 1
Open Review
(x) I would not like to sign my review report
( ) I would like to sign my review report
English language and style
( ) Extensive editing of English language and style required
(x) Moderate English changes required It is re-send for English editor.
( ) English language and style are fine/minor spell check required
( ) I don't feel qualified to judge about the English language and style
Yes | Can be improved | Must be improved | Not applicable | |
Does the introduction provide sufficient background and include all relevant references? | (x) | ( ) | ( ) | ( ) |
*Is the research design appropriate? | ( ) | ( ) | (x) | ( ) |
**Are the methods adequately described? | ( ) | ( ) | (x) | ( ) |
***Are the results clearly presented? | ( ) | ( ) | (x) | ( ) |
****Are the conclusions supported by the results? | ( ) | (x) | ( ) | ( ) |
Comments and Suggestions for Authors
*Is the research design appropriate? | ( ) | ( ) | (x) | ( ) |
Research design is re-structured as:
1. Introduction
2. Objective
3. Literature
4. Methodology
4.1 Dynamic Thermal Simulations and Analysis
4.2 Results and Discussions on Energy Performance
4.3 Results and Discussions on Thermal Comfort
5. Results and Discussions on Energy Performance
6. Results and Discussions on Thermal Comfort
7. Conclusions
**Are the methods adequately described? | ( ) | ( ) | (x) | ( ) |
For the description of method, Figures 3 and 4 added for description of case study building model as plan and 3D with analysis process.
*Figure 3. Case study models, as sample the plan (A) and 3D (B) of the flat office building.
*Figure 4. The research simulation and analysis process.
***Are the results clearly presented? | ( ) | ( ) | (x) | ( ) |
In structure of article new sub chapters are included.
1. Introduction
2. Objective
3. Literature
4. Methodology
4.1 Dynamic Thermal Simulations and Analysis
4.2 Results and Discussions on Energy Performance
4.3 Results and Discussions on Thermal Comfort
5. Results and Discussions on Energy Performance
6. Results and Discussions on Thermal Comfort
7. Conclusions
Now, sub chapter 5 presents results and discussions for energy performance and sub chapter 6 represents results and discussions for thermal comfort.
****Are the conclusions supported by the results? | ( ) | (x) | ( ) | ( ) |
Conclusion is re-written and below paragraph is newly added.
According to energy performance parameters’ as building heat gaining, losing and mean radiant temperature, the dynamic simulation models of the north-west atrium orientation in office building with 0% windows opening ratio had worst yearly average energy performance as about 103.4 W heat losing and gaining and also about 34.4 °C in office and 36.7 °C in atrium mean radiant temperature. However the south-east atrium orientation then north-east atrium orientation in office building with 0% windows opening ratio had negative energy performance behavior. As thermal comfort parameters can be mention that the south-west atrium orientation of office building simulation models in office zone with 25%, 50%, 75% and 100% windows opening ratio in September, October, November and December and in atrium zone from May to October according to predicted mean vote were not been thermal comfort. Furthermore this simulation models group with 25%, 50%, 75% and 100% windows opening ratio in office zone from May to October and atrium zone in September, October and November based on predicted percentage dissatisfied were not been thermal comfort.
I am very sorry to say the following frankly:
To me this is like a dissertation report, without new understanding brought to this field. Dynamic modelling now has been routinely applied to new building design in many countries due to widely accessibility of the programs like TAS.
There is no novelty in application and methodology.
The model needs validation before discuss its prediction results.
No impact to the research communit. Posible good for practitioners or architects and clients who need to be convinced that dynamic thermal modelling is essential to create a good design that provides comfort with less energy.

Reviewer 2 Report
Please subdivide chapters 4.2 and 4.3 into sub-chapters in order to facilitate reading.
Please review the layout of the tables, the top row is not legible (column headings)
Author Response
Reviewer 2
Open Review
(x) I would not like to sign my review report
( ) I would like to sign my review report
English language and style
( ) Extensive editing of English language and style required
( ) Moderate English changes required
(x) English language and style are fine/minor spell check required It is checked by English editor again.
( ) I don't feel qualified to judge about the English language and style
Yes | Can be improved | Must be improved | Not applicable | |
Does the introduction provide sufficient background and include all relevant references? | (x) | ( ) | ( ) | ( ) |
Is the research design appropriate? | (x) | ( ) | ( ) | ( ) |
Are the methods adequately described? | (x) | ( ) | ( ) | ( ) |
Are the results clearly presented? | (x) | ( ) | ( ) | ( ) |
Are the conclusions supported by the results? | (x) | ( ) | ( ) | ( ) |
Comments and Suggestions for Authors
Please subdivide chapters 4.2 and 4.3 into sub-chapters in order to facilitate reading.
In structure of article new sub chapters included.
1. Introduction
2. Objective
3. Literature
4. Methodology
4.1 Dynamic Thermal Simulations and Analysis
4.2 Results and Discussions on Energy Performance
4.3 Results and Discussions on Thermal Comfort
5. Results and Discussions on Energy Performance
6. Results and Discussions on Thermal Comfort
7. Conclusions
Now, sub chapter 5 presents results and discussions for energy performance and sub chapter 6 represents results and discussions for thermal comfort.
Please review the layout of the tables, the top row is not legible (column headings)
Table headings are all re-organized.

Reviewer 3 Report
This paper could be accepted for publication with the following revisions
The article needs to well re-organized and structured especially the first three paragraphs which confuse the readers.I think the first paragraph should be merged and syntetized with the third. In addiction, the authors should consider to re-write the abstract in function the objective of the paper “The aim of this paper is to determine the optimal single-story office building model with a corner atrium type regarding different atrium orientations and office building window opening ratios in the Mediterranean climate. “
Please, the authors must specify the choice of the U-values and properties in the models (i.e. Maximum Allowable U-values in the Latest Building Regulations?) and how the could be influenced the performance of the model
Please, It should be inserted an image of the model
The paragraph "Results and Discussions on Energy Performance" is very tedious. I suggest to Authors to report the most significative results with the help of tables and figures
Please improve the quality of the following pictures/images: Figure 1 - Figure 2
Please improve all tables: the first line is unreadable.
Author Response
Reviewer 3
Open Review
(x) I would not like to sign my review report
( ) I would like to sign my review report
English language and style
( ) Extensive editing of English language and style required
(x) Moderate English changes required It is send to English editor again.
( ) English language and style are fine/minor spell check required
( ) I don't feel qualified to judge about the English language and style
Yes | Can be improved | Must be improved | Not applicable | |
*Does the introduction provide sufficient background and include all relevant references? | ( ) | (x) | ( ) | ( ) |
**Is the research design appropriate? | ( ) | ( ) | (x) | ( ) |
***Are the methods adequately described? | ( ) | ( ) | (x) | ( ) |
****Are the results clearly presented? | ( ) | ( ) | (x) | ( ) |
*****Are the conclusions supported by the results? | ( ) | (x) | ( ) | ( ) |
Comments and Suggestions for Authors
*Does the introduction provide sufficient background and include all relevant references? | ( ) | (x) | ( ) | ( ) |
In introduction part there are 27 strong references. For introduction part new sentence added to the beginning of the chapter.
As important issues of the building for provision of occupant’s comforts in the different seasons, indoor thermal comfort cannot just rely on the passive strategies of cooling. Building sectors usually….
**Is the research design appropriate? | ( ) | ( ) | (x) | ( ) |
Research design is re-structured as:
Introduction
Objective
Literature
Methodology
4.1 Dynamic Thermal Simulations and Analysis
4.2 Results and Discussions on Energy Performance
4.3 Results and Discussions on Thermal Comfort
Results and Discussions on Energy Performance
Results and Discussions on Thermal Comfort
Conclusions
***Are the methods adequately described? | ( ) | ( ) | (x) | ( ) |
For the description of method Figures 3 and 4 added for description of case study building model as plan and 3D with analysis process.
Figure 3. Case study models, as sample the plan (A) and 3D (B) of the flat office building.
Figure 4. The research simulation and analysis process.
****Are the results clearly presented? | ( ) | ( ) | (x) | ( ) |
In structure of article new sub chapters included. Introduction Objective Literature Methodology 4.1 Dynamic Thermal Simulations and Analysis 4.2 Results and Discussions on Energy Performance 4.3 Results and Discussions on Thermal Comfort Results and Discussions on Energy Performance Results and Discussions on Thermal Comfort Conclusions Now, sub chapter 5 presents results and discussions for energy performance and sub chapter 6 represents results and discussions for thermal comfort.
*****Are the conclusions supported by the results? Conclusion is re-written and below paragraph is newly added.
According to energy performance parameters’ as building heat gaining, losing and mean radiant temperature, the dynamic simulation models of the north-west atrium orientation in office building with 0% windows opening ratio had worst yearly average energy performance as about 103.4 W heat losing and gaining and also about 34.4 °C in office and 36.7 °C in atrium mean radiant temperature. However the south-east atrium orientation then north-east atrium orientation in office building with 0% windows opening ratio had negative energy performance behavior. As thermal comfort parameters can be mention that the south-west atrium orientation of office building simulation models in office zone with 25%, 50%, 75% and 100% windows opening ratio in September, October, November and December and in atrium zone from May to October according to predicted mean vote were not been thermal comfort. Furthermore this simulation models group with 25%, 50%, 75% and 100% windows opening ratio in office zone from May to October and atrium zone in September, October and November based on predicted percentage dissatisfied were not been thermal comfort.
|
This paper could be accepted for publication with the following revisions
The article needs to well re-organized and structured especially the first three paragraphs which confuse the readers.I think the first paragraph should be merged and syntetized with the third. First three paragraphs are merged in Introduction.
In addiction, the authors should consider to re-write the abstract in function the objective of the paper “The aim of this paper is to determine the optimal single-story office building model with a corner atrium type regarding different atrium orientations and office building window opening ratios in the Mediterranean climate. “ Abstract is re-written accordingly.
Please, the authors must specify the choice of the U-values and properties in the models (i.e. Maximum Allowable U-values in the Latest Building Regulations?) and how the could be influenced the performance of the model
U-values used mostly for this climate are examined from literature with widely used U-values in market considered for choices made accordingly. Maximum and minimum U values researched. Effect of chosen U-values on the simulated building are put as findings and discussed accordingly.
All of the simulation models of the office building used opaque construction layers with U-values and properties as follows:
Ground floor: U-value of 0.283 (W/m2K), 0.760 external and 0.500 Internal surface of solar absorptance, 0.910 external and 0.900 internal emissivity, 0.297 conductance (W/m2K) and 127.999 time constant.
Ceiling: U-value of 1.01 (W/m2K), 0.700 external and 0.500 internal surface of solar absorptance, 0.900 external and internal emissivity, 1.251 conductance (W/m2K), 13.749 time constant.
All of the simulation models of the office building used glass construction layers with U-values, and properties as follows:
Walls: 1.803 (W/m2K), 0.498 solar transmittance, 0.173 external and 0.135 internal surface of external solar absorptance, 0.227 external and 0.097 internal surface of internal solar absorptance, 0.760 light transmittance, 0 time constant.
Windows (clear 6-12-6 double glazing low E): 1.803 (W/m2K), 0.498 solar transmittance, 0.173 external and internal surface of external solar absorptance, 0.227 external and 0.097 internal surface of internal solar absorptance, 0.760 light transmittance, 0 time constant.
All of the simulation models of the office building used opaque construction layers with U-values and properties as follows:
§ Ground floor: External U-value with horizontal flow direction is 0.283 (W/m2. ⁰C), 0.760 external and 0.500 Internal surface of solar absorptance, 0.910 external and 0.900 internal emissivity (W/m2. ⁰C), 0.297 conductance (W/m2. ⁰C) and 127.999 time constant.
§ Ceiling: External U-value with horizontal flow direction is 1.01 (W/m2. ⁰C), 0.700 external and 0.500 internal surface of solar absorptance, 0.900 external and internal emissivity (W/m2. ⁰C), 1.251 conductance (W/m2. ⁰C), 13.749 time constant.
§ Brick external walls: 229 mm plastered brick wall with horizontal flow direction is 1.135 (W/m2. ⁰C), 0.400 external and internal surface of solar absorptance, 0.900 external and internal emissivity (W/m2. ⁰C), 1.407 conductance (W/m2. ⁰C), 4.920 time constant.
All of the simulation models of the office building used glass construction layers with U-values, and properties as follows:
§ Glass for all windows (clear 6-12-6 double glazing low E): 1.94 (W/m2. ⁰C), 0.498 solar transmittance, 0.550, 0.162 external and 0.157 internal for external solar absorptance, 0.209 external and 0.107 internal for internal solar absorptance, 0.797 light transmittance, 0.845 external and 0.845 internal emissivity, 2.896 conductance (W/m2. ⁰C), 0 for time constant with no blind.
Please, It should be inserted an image of the model
Image of the model inserted as Figure 3.
The paragraph "Results and Discussions on Energy Performance" is very tedious. I suggest to Authors to report the most significative results with the help of tables and figures
In structure of article new sub chapters included.
Introduction
Objective
Literature
Methodology
4.1 Dynamic Thermal Simulations and Analysis
4.2 Results and Discussions on Energy Performance
4.3 Results and Discussions on Thermal Comfort
Results and Discussions on Energy Performance
Results and Discussions on Thermal Comfort
Conclusions
Now, sub chapter 5 presents results and discussions for energy performance and sub chapter 6 represents results and discussions for thermal comfort.
Additionally figure 5 located in this section. (5. Results and Discussions on Energy Performance)
Figure 5. The building heat flow (W) and mean radiant temperature (°C) for the office and atrium zones in different atrium orientation simulation models.
ALL THE FOLLOWING PARAGRAPHS REMOVED FROM,
5. Results and Discussions on Energy Performance
At the 50% window opening ratio for south-east atrium orientation simulation group, the office zone had the maximum heat loss of 1241.9 W at 1 am on the 1st of January and the minimum of 0.01 W at 3 am on the 24th of January. Furthermore, this zone had the 1208.8 W maximum heat gain at 3 am on the 25th of January and the minimum of 0.04 W heat gain at 15 pm on the 27th of July. The atrium zone had the 266.5 W maximum heat loss at 1 am on the 1st of January and the 0 W minimum heat loss at 2 pm on the 8th of August. The atrium zone also had the maximum heat gain of 500.8 W at 5 am on the 19th of July and the 0 W minimum heat gain at 2 pm on the 8th of August. At the 75% window opening ratio with the same atrium orientation, the office zone had the 1259.3 W maximum heat loss at 1 am on the 1st of January and the minimum heat loss of 0.04 W at 6 am on the 5th of January. Additionally, the maximum heat gain of the office zone was 1250.1 W at 3 am on the 25th of January, and the minimum was 0.01 W heat gain at 19 pm on the 29th of April. The atrium zone of this group had the 258.3 W maximum heat loss at 1 am on the 1st of January and the 0.02 W minimum heat loss value at 15 pm on the 28th of July. Moreover, the atrium zone had the 476.9 W maximum heat gain at 6 pm on the 4th of April and the minimum of 0.2 W heat gain at 2 pm on the 12th of April. When all of the windows were opened completely (100% opening), the office zone had the 1274.9 W maximum heat loss at 1 am on the 1st of January, and its minimum was 0.01 W heat loss at 5 am on the 2nd of May. Additionally, it had the 1266.5 W maximum heat gain at 3 am on the 25th of January and the minimum of 0.04 W heat gain at 4 pm on the 3rd of October. The atrium zone with the same simulation parameters had the 199.9 W maximum heat loss at 11 pm on the 25th of November and the minimum 0.03 W heat loss at 5 pm on the 17th of September. Furthermore, this zone had the 634.2 W maximum heat gain at 4 pm on the 15th of January and the minimum of 0.03 W heat gain at 6 pm on the 21st of January.
For the 50% window opening ratio in the north-east atrium orientation simulation group, the office zone had the 1286 W maximum heat loss value at 1 am on the 1st of January, and its minimum was 0.06 W heat loss at 11 pm on the 9th of October. Furthermore, the office zone had the 1186 W maximum heat gain at 3 am on the 25th of January, and its minimum was 0.04 W heat gain at 3 pm on the 24th of October. The atrium zone had the 222.9 W maximum heat loss at 1 am on the 1st of January, and its minimum was 0.08 W heat loss at 7 pm on the 2nd of February. This space also had the 541.8 W maximum heat gain at 7 am on the 14th of February and the minimum heat gain of 0.01 W at 3 pm on the 7th of April. At the 75% window opening ratio for the same atrium orientation, the office zone had the 1391 W maximum heat loss at 7 am on the 25th of January, and its minimum was 0.02 W heat loss at 5 pm on the 12th of September. Additionally, the office space had the 1226.2 W maximum heat gain at 3 am on the 25th of January, and its minimum was 0.01 W heat gain at 6 am on the 29th of May. The atrium zone had the 223 W maximum heat loss at 1 am on the 1st of January and the minimum heat loss of 0.06 W at 9 pm on the 30th of October. Additionally, this zone had the maximum heat gain of 519.2 W at 10 am on the 23rd of January and its minimum heat gain as 0.02 W at 6 pm on the 9th of February. Furthermore, when all of the windows were opened completely (100% opening), the office zone had the 1460.7 W maximum heat loss value at 7 am on the 25th of January, and its minimum was 0.01 W heat loss at 5 pm on the 28th of November. This zone had the 1271.8 W maximum heat gain at 6 am on the 5th of April and the minimum heat gain of 0.03 W at 12 am on the 30th of July. The atrium zone of this group simulation had the 219.7 W maximum heat loss at 1 am on the 1st of January, and the minimum was 0.01 W heat loss at 11 pm on the 9th of March. Additionally, the atrium zone had the 508.7 W maximum heat gain value at 10 am on the 23rd of January, and its minimum was 0.02 W heat gain at 10 pm on the 29th of October.
The office building simulation models with different atrium orientations and the 0% window opening ratio had the same mean radiant temperature (MRT), but whenever the window opening ratio increased, the MRT of the south-east, north-west and north-east orientations slightly decreased. However, the south-west atrium orientation simulation model still had the same MRT from 0% up to the 100% window opening ratio. The office building simulation model of the south-east atrium orientation with the 0% window opening ratio had a yearly mean radiant temperature average of 36.72 °C. The simulation model with the 100% window opening ratio had its yearly MRT average as 29.88 °C, while the MRT of the north-east atrium orientation model was approximately 28 °C. The north-west atrium orientation with the 0% window opening ratio had a yearly mean radiant temperature of 34.46 °C in the office zone and 36.72 °C in the atrium zone. In contrast, the 100% window opening ratio had its mean radiant temperature (MRT) as 29.88 °C and 30.67 °C for the office and atrium zones respectively.
Throughout May, June, July, and August, the office zone in the north-west atrium orientation simulation models when all the windows were closed (0% opening) had from 211.4 W up to 242.5 W heat loss and 282.8 W up to 364.7 W heat gain; the ones with the 25% window opening ratio had 163.3 W up to 167.4 W heat loss and 160.1 W up to 224.1 W heat gain; the ones with the 50% window opening ratio had 142.1 W up to 163.3 W heat loss and 159 W up to 216.7 W heat gain; the ones with the 75% window opening ratio had 140.2 W up to 161.9 W heat loss and 155.2 W up to 214.4 W heat gain, and the ones with the 100% window opening ratio had 141.1 W up to 164.5 W heat loss and 153.6 W up to 216.7 W heat gain. Furthermore, the atrium zone in the same months, with a 0% window opening ratio had from 20.7 W up to 26 W heat loss and 189.7 W up to 241.3 W heat gain; the ones with the 25% window opening ratio had 44.1 W up to 60.2 W heat loss and 138.9 W up to 147.7 W heat gain; the ones with the 50% window opening ratio had 41.7 W up to 54.9 W heat loss and 129.7 W up to 143.4 W heat gain; the ones with the 75% window opening ratio had 40.3 W up to 52.3 W heat loss and 125.4 W up to 143 W heat gain, and the ones with the 100% window opening ratio had 39.2 W up to 49.9 W heat loss and 122.2 W up to 147.1 W heat gain.
Throughout September, October, November and December, the office zone in the dynamic simulation model with a south-west atrium orientation when the windows were completely closed (0% opening) had from 179.5 W up to 194.4 W heat loss and 189.1 W up to 222.3 W heat gain; the ones with the 25% window opening ratio had 180.3 W up to 194.4 W heat loss and 188.9 W up to 223.2 W heat gain; and the ones with the 50%, 75% and 100% window opening ratios had from 180.3 W up to 194.4 W heat loss and 188.9 W up to 223.2 W heat gain. The remarkable achievement of the atrium zone in this group was that the building heat transfer in these months with the different window opening ratios (0%, 25%, 50%, 75%, and 100%) was still same with between 86.8 W and 129 W heat loss, and 49.5 W and 167.8 W heat gain. The office zone of the south-east atrium orientation in the simulation models and a 0% window opening ratio had from 105.3 W up to 133.3 W heat loss and 98.3 W up to 145.4 W heat gain; the ones with the 25% window opening ratio had 95.7 W up to 108.3 W heat loss and 143.8 W up to 172.1 W heat gain; the ones with the 50% window opening ratio had 93.7 W up to 103.6 W heat loss and 140.9 W up to 173 W heat gain; the ones with the 75% window opening ratio had 94.6 W up to 109.7 W heat loss and 139.6 W up to 172.5 W heat gain, and the ones with the 100% window opening ratio had 93.4 W up to 109.5 W heat loss and 141.3 W up to 174.6 W heat gain. The atrium zone in this simulation group when all windows were closed (0% opening) had from 91.9 W up to 124.5 W heat loss and 161.3 W up to 224 W heat gain; the ones with the 25% window opening ratio had 62.2 W up to 72.9 W heat loss and 91 W up to 140.7 W heat gain; the ones with the 50% window opening ratio had 64.5 W up to 70.8 W heat loss and 81.8 W up to 129.4 W heat gain; the ones with the 75% window opening ratio had 65.6 W up to 69.7 W heat loss and 80.5 W up to 124.5 W heat gain; and when all windows were opened completely (100% opening), it had from 64.6 W up to 69.1 W heat loss and 75.7 W up to 126.3 W heat gain.
Please improve the quality of the following pictures/images: Figure 1 - Figure 2
Both figures 1 and 2 improved.
Please improve all tables: the first line is unreadable.
All table headings are made readable. All Tables are improved accordingly.

Round 2
Reviewer 1 Report
sustainability-426271
The Abstract
This is very well written, very clear and informative summarises well the entire study. The unit showing energy performance is not appropriate. It should use kWh/m2
The introduction
The introduction need much improvement. The background should provide info that is relevant to this study – to convince reader that your research is indeed needed to be done. The evidence and literature should be given so that we can see the study would have an impact to the science community. Current such justification for this study is not built.
The Methods
I am afraid that I am very disappointed when I have finished reading this section. It is very good report for a modelling study carried out to support the design team to make sound decision for specific project in the region. But not a PEER Reviewed Research paper and a international journal.
First the authors should have a section describing weather features especially in the summer so the reader can appreciate the major and possible causes for indoor discomfort and then you research outcomes. This can also help readers to relate the climate to others in the referred studies, such as Santiago.
Second is about the conceptual building made for modelling. How representative of this conceptual building is? How the results of this building could be used in other cases in the region? The journal is an international journal, you need to convince that the report is providing useful information for the majority of the readers to justify publication. Double glazed windows in Mediterranean buildings? Double glazed windows is normally for preventing heat losses. In Cyprus, does building needs heating in winters? Or it is the summer preventing over heating and solar penetration more of a problem.
Third, many Mediterranean buildings have removable shutters or blinds to control summer solar penetration through their windows. A study by F Wang, K Pichatwatana, S Roaf, L Zhao, Z Zhu, J Li, titled “Developing a weather responsive internal shading system for atrium spaces of a commercial building in tropical climates” (Building and Environment V71, 259-274, 2014) has examined the effects the status an operable shading device on energy and comfort. This should be included in this study. An atrium without shading in hot climate would be a problematic project.
The forth one: the predicted results should be compared against a baseline – the thermal comfort and energy consumption in a normal and common building of the same use in the region. This baseline should come from local governmental authority set as a reference or if Cyprus does not have one (I do not believe so) the authors should establish one for comparison. In the UK, we have “best practice”; “national average” “poor performance” and so on.
The last one which is critical! The paper does not talk about the model’s validation and test. How can we trust the robustness of this model. How can we believe the model is sensitive enough to reveal the difference in the comfort indices and MRT prediction for the window opening ratios?
Author Response
Reviewer 1
Open Review
(x) I would not like to sign my review report
( ) I would like to sign my review report
English language and style
( ) Extensive editing of English language and style required
(x) Moderate English changes required. English editing is done.
( ) English language and style are fine/minor spell check required
( ) I don't feel qualified to judge about the English language and style
Yes | Can be improved | Must be improved | Not applicable | |
*Does the introduction provide sufficient background and include all relevant references? | ( ) | ( ) | (x) | ( ) |
**Is the research design appropriate? | ( ) | ( ) | (x) | ( ) |
***Are the methods adequately described? | ( ) | (x) | ( ) | ( ) |
****Are the results clearly presented? | ( ) | (x) | ( ) | ( ) |
*****Are the conclusions supported by the results? | ( ) | (x) | ( ) | ( ) |
*Introduction part is improved by adding references 1, 5 and 17.
**Research design is improved by adding references 48 and 49.
***Method is developed by adding information about Gazimagusa weather conditions and references made to all figures and tables accordingly. All of them are in the first paragraph of Methodology.
****Figures 6-21 shows base horizontal lines for categories with Tables showing which month thermal comfort generated according to which baseline or category.
*****Conclusion is improved by adding last paragraph.
Comments and Suggestions for Authors
sustainability-426271
The Abstract
This is very well written, very clear and informative summarises well the entire study. The unit showing energy performance is not appropriate. It should use kWh/m2:
In this article Watt (W) is used for the heat loss/gains that TAS software is using also.
The introduction
The introduction need much improvement. The background should provide info that is relevant to this study – to convince reader that your research is indeed needed to be done. The evidence and literature should be given so that we can see the study would have an impact to the science community. Current such justification for this study is not built.
Introduction with background parts are improved by adding reference 1, 5, 17.
Literature review is improved by adding references 48 and 49.
The Methods
I am afraid that I am very disappointed when I have finished reading this section. It is very good report for a modelling study carried out to support the design team to make sound decision for specific project in the region. But not a PEER Reviewed Research paper and a international journal.
First the authors should have a section describing weather features especially in the summer so the reader can appreciate the major and possible causes for indoor discomfort and then you research outcomes. This can also help readers to relate the climate to others in the referred studies, such as Santiago.
Weather of Gazimagusa explained in first paragraph of methodology section by reference 51. The paragraph continued with referring to figures-tables for comparing minimum and maximum heat losses, heat gains, PMV, PPD for office and atrium with their comparatively performances according to different categories.
Second is about the conceptual building made for modelling. How representative of this conceptual building is? The model shape chosen because of popularity of designing here. How the results of this building could be used in other cases in the region? This is the aim of choosing most popular simple geometry for generation of performances in comparative way, when most heat loss/gain, draught etc. happens which month occurs.. can be used in other cases also. The journal is an international journal, you need to convince that the report is providing useful information for the majority of the readers to justify publication. Double glazed windows in Mediterranean buildings? Double glazed windows is normally for preventing heat losses. In Cyprus, does building needs heating in winters? Or it is the summer preventing over heating and solar penetration more of a problem. This type of window is most popular one. Using the double glazed windows used because during cold seasons, building needs heating, and also during warm seasons used for preventing the heat transfer.
Third, many Mediterranean buildings have removable shutters or blinds to control summer solar penetration through their windows. A study by F Wang, K Pichatwatana, S Roaf, L Zhao, Z Zhu, J Li, titled “Developing a weather responsive internal shading system for atrium spaces of a commercial building in tropical climates” (Building and Environment V71, 259-274, 2014) has examined the effects the status an operable shading device on energy and comfort. This should be included in this study. An atrium without shading in hot climate would be a problematic project.
This reference is added to literature review part as reference 49 to this study.
The reason for simulating all of the models without any shading devices is assessing the performance of the office building as energy performance and internal thermal comfort (popular approach here). So, having this kind of findings can be suitable for designing a building based on the architectural parameters according the local climate, as an atrium orientation into a flat building (common pattern in Gazimagusa) and all windows opening ratio, then as the next stage can decide and use the shading devices as a solution and developing internal comfort.
The forth one: the predicted results should be compared against a baseline – the thermal comfort and energy consumption in a normal and common building of the same use in the region. This baseline should come from local governmental authority set as a reference or if Cyprus does not have one (I do not believe so) the authors should establish one for comparison. In the UK, we have “best practice”; “national average” “poor performance” and so on. In this research all of the data analyzed based on the referenced standards as the ISO 7730: 2005 and EN 15251: 2007 standards.
Our base line is given in Figure 6-21, in comparative way for each month etc. Tables 1-6 shows when thermal comfort is generated according to our base line or categories for draught etc.
The last one which is critical! The paper does not talk about the model’s validation and test. How can we trust the robustness of this model. How can we believe the model is sensitive enough to reveal the difference in the comfort indices and MRT prediction for the window opening ratios?
The study is only covers dynamic thermal simulations by TAS software. Reading of examples from referances 15, 16, 21 and 22, the aim of this study to generate monthly thermal comfort issues for the readers.

Reviewer 3 Report
Accept in present form
Author Response
Reviewer 3
Open Review
(x) I would not like to sign my review report
( ) I would like to sign my review report
English language and style
( ) Extensive editing of English language and style required
( ) Moderate English changes required.
(x)English language and style are fine/minor spell check required English editing is done.
( ) I don't feel qualified to judge about the English language and style
Yes | Can be improved | Must be improved | Not applicable | |
Does the introduction provide sufficient background and include all relevant references? | (x) | ( ) | () | ( ) |
Is the research design appropriate? | (x) | ( ) | () | ( ) |
Are the methods adequately described? | (x) | () | ( ) | ( ) |
Are the results clearly presented? | (x) | () | ( ) | ( ) |
Are the conclusions supported by the results? | (x) | () | ( ) | ( ) |

Round 3
Reviewer 1 Report
It is good to add a summery of the local weather condition. A charge would be more effective for the same purpose if the monthly and daily everage maximum, mean and minimum were provided for each month in a year. This is a standard way. It should be very easy for the authours to do so and very easy too for all readers to have a full picture of the local weather conditions.
More importantly the daily everage maximum, mean and minimum would show the diaurnal temperatures that would allow designers to explore the possibility of night ventilation for better free cooling for office and commercial buildings when they are not occupied during the night.
This actually raises another proble, which is a fundamental issue that is not clear in the paper. The authors point out that "the window opening ratio would be based on the outside air temperature, time, the user’s pattern and season" and then conclude correctly that it "is an extremely important factor in regards to the users’ indoor comfort and energy consumption". In reality, it important indeed how much it should be opened and it is even more important when it should be open.
The paper does not clearly explore when the windows were modelled open. For a single storey office, it would be very problematic to allow all windows open during the nights for security reasons. It would also very unrealistic to have the windows fully open in cold seasons, and fully closed in hot months. Without clear information on the open schedule and how this modelled, the findings do not carry more useful information for designers than common sense.
The model validation issue has not be addressed. A computer model without validation or any sensitivity test procedure should not be used for an investigation. What do you show your readers that your model is robust or your model is sensitive enough to reveal correctly the effect of the changes of the selected design variables. The revised paper still do not provide convincing explanation for this point.